# MATÉRN KERNELS FOR TUNABLE IMPLICIT SURFACE RECONSTRUCTION

**Maximilian Weiherer & Bernhard Egger**
Department of Computer Science
Friedrich-Alexander-Universität Erlangen-Nürnberg
`{maximilian.weiherer,bernhard.egger}@fau.de`

## ABSTRACT

We propose to use the family of Matérn kernels for implicit surface reconstruction, building upon the recent success of kernel methods for 3D reconstruction of oriented point clouds. As we show from a theoretical and practical perspective, Matérn kernels have some appealing properties which make them particularly well suited for surface reconstruction—outperforming state-of-the-art methods based on the *arc-cosine* kernel while being significantly easier to implement, faster to compute, and scalable. Being stationary, we demonstrate that Matérn kernels allow for *tunable* surface reconstruction in the same way as Fourier feature mappings help coordinate-based MLPs overcome spectral bias. Moreover, we theoretically analyze Matérn kernels' connection to SIREN networks as well as their relation to previously employed arc-cosine kernels. Finally, based on recently introduced Neural Kernel Fields, we present *data-dependent* Matérn kernels and conclude that especially the Laplace kernel (being part of the Matérn family) is extremely competitive, performing almost on par with state-of-the-art methods in the noise-free case while having a more than five times shorter training time.[1]

## 1 INTRODUCTION

Recovering the shape of objects from sparse or only partial observations is a challenging task. Formally, let $\Omega \subset \mathbb{R}^d$ and $\mathcal{X} = \{x_1, x_2, \ldots, x_m\} \subset \Omega$ be a set of $m$ points in $d$ dimensions which forms, together with associated per-point normals, a dataset $\mathcal{D} = \mathcal{X} \times \{n_1, n_2, \ldots, n_m\} \subset \Omega \times \mathbb{R}^d$. The goal of surface reconstruction is to recover the object's shape from $\mathcal{D}$. The shape of objects may be represented as dense point clouds, polygonal meshes, manifold atlases, voxel grids, or (as the zero-level set of) implicit functions, which is the representation we chose to focus on in this work. Specifically, if $f : \mathbb{R}^d \longrightarrow \mathbb{R}$ denotes an implicit function, such as a *Signed Distance Function* (SDF), then, from a practical perspective, implicit surface reconstruction aims at finding an optimal solution $\hat{f}$ to the following *Kernel Ridge Regression* (KRR) problem:

$$\hat{f} = \arg\min_{f \in H} \left\{ \sum_{i=1}^{m} |f(x_i)|^2 + \|\nabla f(x_i) - n_i\|^2 + \lambda \|f\|_H^2 \right\}, \tag{1}$$

where the function space $H := H(\Omega)$ is usually taken to be a *Reproducing Kernel Hilbert Space* (RKHS) with associated reproducing kernel $k : \Omega \times \Omega \longrightarrow \mathbb{R}$, and $\lambda > 0$. Once $\hat{f}$ at hand, the reconstructed object's surface is implicitly given as $\hat{\mathcal{S}} = \{x : \hat{f}(x) = 0\} \subset \mathbb{R}^d$ and can be extracted using Marching Cubes (Lorensen & Cline, 1987). It is quite easy to observe that, as $\lambda \to 0$, the optimization problem in Eq. (1) turns into the constrained minimization problem,

$$\underset{f \in H}{\text{minimize}} \|f\|_H \quad \text{s.t.} \quad f(x_i) = 0 \quad \text{and} \quad \nabla f(x_i) = n_i \quad \forall i \in \{1, 2, \ldots, m\} \tag{2}$$

which makes it easier to see how the predicted surface behaves *away* from the input points. As seen from Eq. (2), while $\hat{\mathcal{S}}$ should interpolate the given points $\mathcal{X}$ exactly, the behavior in between and away from those points is solely determined by the induced norm $\|f\|_H$ of the function space, $H$, over which we are optimizing. The properties of $\|f\|_H$ are uniquely defined by

---

[1]Code available at: `https://github.com/mweiherer/matern-surface-reconstruction`.

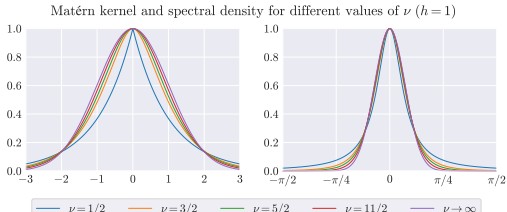
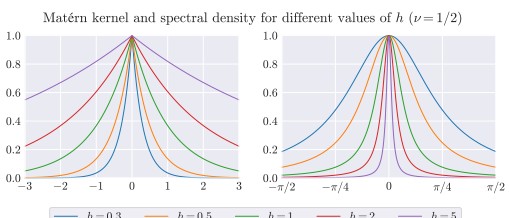

Figure 1: **Matérn kernels with associated spectral densities.** We propose to use the family of Matérn kernels for tunable implicit surface reconstruction, parametrized by a smoothness parameter, $\nu > 0$, that controls the differentiability of the kernel, and a bandwidth parameter $h > 0$. Both parameters allow explicit manipulation of the kernel and its spectrum. Importantly, Matérn kernels recover the Laplace kernel for $\nu = 1/2$ and the Gaussian kernel as $\nu \to \infty$.

the reproducing kernel, $k$; this is a direct consequence of the Representer Theorem (Kimeldorf & Wahba, 1970; Schölkopf et al., 2001) which states that solutions to Eq. (1) are of the form $f(x) = \sum_{i=1}^{m} \alpha_i k(x, x_i)$, yielding $\|f\|_H^2 = \alpha^\top K \alpha$. Here, $\alpha \in \mathbb{R}^m$ and $K = (K)_{ij} = k(x_i, x_j) \in \mathbb{R}^{m \times m}$. This shows that the behavior of the norm and hence, estimated surfaces $\hat{S}$, can be explicitly controlled by the kernel function, enabling easy injection of *inductive biases* (such as smoothness assumptions) into the surface reconstruction problem. Even further, if the chosen kernel has adjustable parameters, they can be used to *adaptively* (on a shape-by-shape basis) manipulate the inductive bias.

Although already used in the mid-90s and early 2000s (Savchenko et al., 1995; Carr et al., 1997; 2001), only recently, kernel-based methods for 3D implicit surface reconstruction became extremely competitive, with *Neural Kernel Surface Reconstruction* (NKSR; Huang et al. (2023)) eventually evolving as the new state-of-the-art. While early works mostly focused on (polyharmonic) radial basis functions such as thin-plate splines, recent work (Williams et al., 2021; 2022) employ the first-order *arc-cosine* kernel. Introduced by Cho & Saul (2009), arc-cosine kernels have been shown to mimic the computation of (two-layer) fully-connected ReLU networks; indeed, if the layer-width tends to infinity, the first-order arc-cosine kernel is identified with the *Neural Tangent Kernel* (NTK; Jacot et al. (2018)) of the network. One important aspect of this kernel is that it does not have adjustable parameters. While this can be advantageous in some situations, it also prevents users from *tuning* their reconstructions. If the arc-cosine kernel fails to accurately recover a surface, then there is no possibility (except for using more observations) to improve the result.

In this work, we suggest a different family of kernels (with parameters $\nu, h > 0$) for implicit surface reconstruction: Matérn kernels (Matérn, 1986; Stein, 1999), see Figures 1 and 2. Contrary to arc-cosine kernels, Matérn kernels are stationary (hence translation invariant), spatially decaying (thus leading to sparse Gram matrices), and unify a variety of popular kernel functions, including the Laplace and Gaussian kernel. As we will show, Matérn kernels have appealing properties, making them the ideal candidate for kernel-based surface reconstruction. From a practical perspective, we demonstrate that a simple change of the kernel function from arc-cosine to Matérn leads to a consistently improved reconstruction accuracy and a *significant* speedup. From a theoretical perspective, we argue that Matérn kernels allow for *tunable* surface reconstruction in the same way as Fourier feature mappings (Tancik et al., 2020) enable coordinate-based *Multi-layer Perceptrons* (MLPs) to learn high-frequency functions in low-dimensional do-

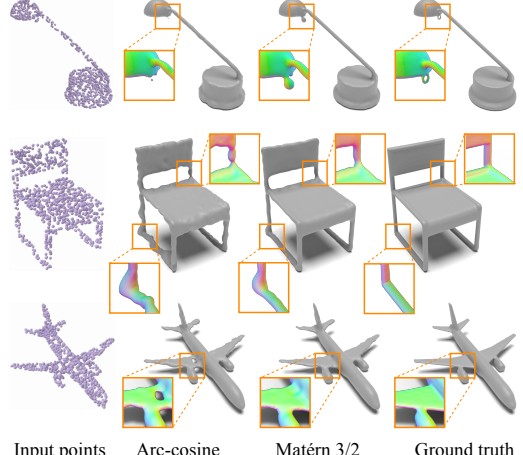

Input points    Arc-cosine    Matérn 3/2    Ground truth

Figure 2: **Tunable Matérn kernels lead to better surface reconstructions than the previously employed arc-cosine kernel**. Here, we show surface reconstructions from sparse point clouds of just 1,000 points.

mains, effectively overcoming *spectral bias* (the slow convergence towards high-frequency details). In addition, we prove that, as the layer-width approaches infinity, Matérn kernels are identified with the NTK of two-layer SIREN (fully-connected MLPs with sine activations; Sitzmann et al. (2020)) networks—together with Fourier feature mappings the two arguably most influential methods to overcome spectral bias in coordinate-based MLPs. Lastly, we establish a connection between arc-cosine and Matérn kernels by showing that the Laplace kernel (Matérn kernel with $\nu = 1/2$) and the arc-cosine kernel are equal up to an affine transformation when restricted to the sphere. In summary, our key contributions are:

- We propose to use Matérn kernels for *tunable* implicit surface reconstruction.
- We theoretically analyze Matérn kernels, relating them to Fourier feature mappings, SIREN networks, and arc-cosine kernels. Moreover, we derive practical insights into how to choose the tunable bandwidth parameter based on a new bound of the $L_2$ reconstruction error.
- We propose data-dependent (*i.e.*, learnable) Matérn kernels by leveraging the *Neural Kernel Field* (NKF) framework (Williams et al., 2022).

Our experimental evaluation reveals that Matérn 1/2 and 3/2 are extremely competitive, outperforming the arc-cosine kernel while being significantly easier to implement (essentially two lines of standard PyTorch code), faster to compute, and scalable. In addition to geometry, we show that Matérn kernels surpass the arc-cosine kernel in reconstructing other high-frequency scene attributes, such as texture. Finally, we demonstrate that learnable Matérn kernels (1) outperform the data-dependent arc-cosine kernel (as implemented in the original NKF framework) while being more than four times faster to train, and (2) perform almost on par with highly sophisticated and well-engineered NKSR in the noise-free case while having a more than five times shorter training time.

## 2 RELATED WORK

We briefly review some relevant literature about 3D *implicit* surface reconstruction from oriented point clouds, focusing on kernel-based methods. For a more in-depth overview, including neural-network-based reconstruction methods, see surveys by Berger et al. (2017); Huang et al. (2022b).

Early kernel-based 3D surface reconstruction methods mostly employ *Radial Basis Functions* (RBFs) such as thin-plate splines (Savchenko et al., 1995), biharmonic RBFs (Carr et al., 1997; 2001), or Gaussian kernels (Schölkopf et al., 2005). Nowadays, the most widely used surface reconstruction technique is *Screened Poisson Surface Reconstruction* (SPSR; Kazhdan & Hoppe (2013)) which, however, can itself be viewed as kernel method (Williams et al., 2021). Only recently, *Neural Splines* (NS; Williams et al. (2021)) proposed to use the so-called (first-order) *arc-cosine* kernel,

$$k_{\text{AC}}(x, y) = \frac{\|x\|\|y\|}{\pi} \left( \sin\theta + (\pi - \theta)\cos\theta \right), \quad \text{where} \quad \theta = \cos^{-1}\left(\frac{x^\top y}{\|x\|\|y\|}\right) \qquad (3)$$

for implicit surface reconstruction, achieving state-of-the-art results that outperform classical surface reconstruction techniques and non-linear methods based on neural networks by a large margin. This method laid the cornerstone for *Neural Kernel Fields* (NKFs; Williams et al. (2022)) which attempts to make arc-cosine kernels learnable by passing input points through a task-specific neural network before evaluating the kernel function, similar to Deep Kernel Learning (Wilson et al., 2016). Based on SPSR, NeuralGalerkin (Huang et al., 2022a) proposed to *learn* basis functions inferred by a sparse convolutional network instead of using a fixed Bézier basis as in SPSR, hence can be seen as kernel method in the broader sense. *Neural Kernel Surface Reconstruction* (NKSR; Huang et al. (2023)) built upon NKF and NeuralGalerkin and proposed an all-purpose, highly scalable surface reconstruction method that is robust against noise and eventually became state-of-the-art.

## 3 MATÉRN KERNELS FOR TUNABLE SURFACE RECONSTRUCTION

We propose to use the family of *Matérn* kernels (Matérn, 1986; Stein, 1999) for implicit surface reconstruction, being defined as

$$k_\nu(x, y) = \Phi_\nu(\|x - y\|) = \Phi_\nu(\tau) = \frac{2^{1-\nu}}{\Gamma(\nu)} \left(\frac{\sqrt{2\nu}\tau}{h}\right)^\nu K_\nu\left(\frac{\sqrt{2\nu}\tau}{h}\right), \qquad (4)$$

where $\nu > 0$ is a *smoothness parameter* that explicitly controls the differentiability, and $h > 0$ is known as the shape parameter (or *bandwidth*) of the kernel. $\Gamma$ denotes the Gamma function, and $K_\nu$ is the modified Bessel function of the second kind of order $\nu$. Matérn kernels generalize a variety of other kernel functions; the most popular ones can be written in closed form as

$$\nu = 1/2 : \Phi_{1/2}(\tau) = \exp\left(-\frac{\tau}{h}\right), \tag{5}$$

$$\nu = 3/2 : \Phi_{3/2}(\tau) = \exp\left(-\frac{\sqrt{3}\tau}{h}\right)\left(1 + \frac{\sqrt{3}\tau}{h}\right), \tag{6}$$

$$\nu = 5/2 : \Phi_{5/2}(\tau) = \exp\left(-\frac{\sqrt{5}\tau}{h}\right)\left(1 + \frac{\sqrt{5}\tau}{h} + \frac{5\tau^2}{3h^2}\right), \tag{7}$$

where $\Phi_{1/2}$ is known as the Laplace kernel. In the limiting case of $\nu \to \infty$, Matérn kernels recover the popular Gaussian kernel:

$$\Phi_\infty(\tau) := \lim_{\nu \to \infty} \Phi_\nu(\tau) = \exp\left(-\frac{\tau^2}{2h^2}\right). \tag{8}$$

For more information about Matérn kernels, please refer to Porcu et al. (2023). Next, we revisit some basic properties of Matérn kernels and place them in the context of surface reconstruction.

### 3.1 BASIC PROPERTIES

**Differentiable.** Matérn kernels allow for *controlled* smoothness, being exactly $\lceil \nu \rceil - 1$ times differentiable (in the mean-square sense). Since functions $f$ in an RKHS $H$ inherit the differentiability class of the inducing kernel $k$ due to the reproducing property, the from Eq. (1) reconstructed surface $\hat{S}$ enjoys the same smoothness properties as $k$. In the context of 3D surface reconstruction, this allows an easy injection of inductive biases into the reconstruction problem; for instance, if the roughness or noisiness of the objects to be reconstructed is known in advance, one can adjust the smoothness of the kernel accordingly. This is not possible with the arc-cosine kernel.

**Stationary.** Matérn kernels are stationary, *i.e.*, they only depend on the difference $x - y$ between two points $x, y \in \Omega$. If the distance is Euclidean, Matérn kernels are also *isotropic*; they only depend on $\|x - y\|$. As a result, Matérn kernels are rotation *and* translation invariant, hence being independent of the absolute embedding of points $\mathcal{X}$ (based on the reproducing property, translating input points does not change the reconstructed surface). Consequently, since the kernel's value only depends on the *relative* positions, objects in a scene (or parts of an object) with similar geometric properties will be reconstructed consistently as long as the relative distances between points on each instance (or part) remain the same. This is in contrast to the arc-cosine kernel, whose value depends on the *absolute* position of points, see Eq. (3); hence, it is *not* translation invariant (thus *non-stationary*), and multiple instances of an object in a scene will generally not yield the same surface reconstruction. See Appendix A for further discussion.

**Spatially decaying.** As opposed to arc-cosine kernels, Matérn kernels are spatially decaying; they tend to zero as the distance $\|x - y\|$ becomes large. Consequently, although Matérn kernels are technically not compact (or locally supported), because their value decays exponentially fast (Porcu et al., 2023), they can be considered *effectively compact*. From a computational perspective, this is a very desirable property as it leads to *sparse* Gram matrices when truncated with a special kernel (Genton, 2001), allowing the use of highly efficient and scalable sparse linear solvers. Finally, we note that simply truncating any kernel does not yield a valid, positive definite kernel in general.

We now derive new theoretical insights into Matérn kernels, ultimately aiming to provide arguments as to why we believe this family of kernels is particularly well suited for surface reconstruction.

### 3.2 EIGENVALUE DECAY, SPECTRAL BIAS, AND RECONSTRUCTION ERROR

**Eigenvalue decay and spectral bias.** We begin by showing that Matérn kernels allow for tunable surface reconstruction in the same way as Fourier feature mappings help coordinate-based MLPs overcome spectral bias. As investigated in Tancik et al. (2020), a rapid decrease of the NTK's eigenvalues implies an associated MLP's slow convergence to high-frequency components of the target

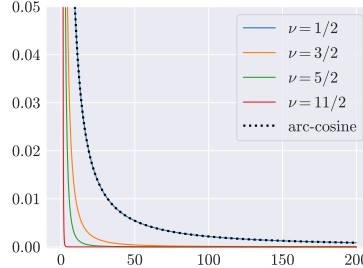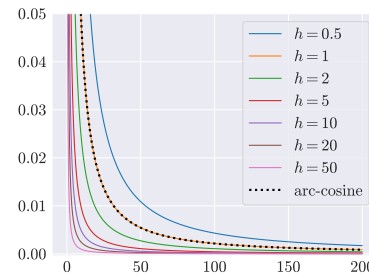

Figure 3: **Eigenvalue decay of Matérn kernels.** While we fix $h = 1$ and vary $\nu$ on the left, the EDR for $\nu = 1/2$ and different values of $h$ is shown on the right. Larger values of $\nu$ and $h$, *i.e.*, smoother kernels, lead to faster eigenvalue decay; hence, slow convergence to high-frequency details.

function (up to the point where the network is practically unable to learn these components). Consequently, a smaller eigenvalue decay rate (EDR), *i.e.*, a slower eigenvalue decay, leads to faster convergence to high-frequency content—in the context of implicit surface reconstruction, more detailed geometry. To overcome this slow convergence, a phenomenon known as *spectral bias*, Tancik et al. (2020) use a Fourier feature mapping of the form

$$\gamma(x) = [a_1 \cos(2\pi b_1^\top x), a_1 \sin(2\pi b_1^\top x), \ldots, a_q \cos(2\pi b_q^\top x), a_q \sin(2\pi b_q^\top x)]^\top \tag{9}$$

applied to the inputs $x \in \mathbb{R}^d$ before passing them to the MLP, effectively transforming the MLP's NTK, $k_{\text{NTK}}$, into a *stationary* kernel, $k_{\text{NTK}}(\gamma(x)^\top \gamma(y)) = k_{\text{NTK}}(k_\gamma(x - y)) =: k'_{\text{NTK}}(x - y)$ whose spectrum can be tuned through manipulation of the Fourier basis frequencies, $b_i \in \mathbb{R}^d$, and corresponding coefficients, $a_i \in \mathbb{R}$, of the kernel $k_\gamma(x - y) = \sum_{i=1}^q a_i^2 \cos(2\pi b_i^\top (x - y))$. This shows that $\gamma$ enables explicit control over the NTK's EDR, overcoming spectral bias.

Matérn kernels, being stationary, allow for the same degree of control over the EDR as a Fourier feature mapping; their spectrum can also be tuned (by varying $\nu$ and/or $h$, see Figure 1). This is in stark contrast to previously employed, non-stationary and parameter-less arc-cosine kernels, whose spectrum is not tunable. To further study the dependence of the EDR on $\nu$ and $h$, we use:

**Theorem 1** (Seeger et al. (2008), Theorem 3). *The eigenvalues of Matérn kernels decay polynomially at rate*

$$\Theta\left(h^{-2\nu} s^{-(1+2\nu/d)}\right)$$

*with bandwidth parameter $h > 0$, and for* finite *smoothness $0 < \nu < \infty$.*

Figure 3 visualizes the EDR for different values of $\nu$ and $h$ along with the EDR of the non-tunable, first-order arc-cosine kernel which equates to $\Theta(s^{-(1+d)/d})$ and has only been very recently proven by Li et al. (2024) for general input domains and distributions. As seen, the smoother the kernel (*i.e.*, the larger $\nu$) and the greater $h$, the faster the eigenvalues decay. This implies less detailed surface reconstruction (with a higher error) for smoother kernels and larger $h$; indeed, this is exactly what we observe in practice, see Figure 4. Contrary, by lowering both parameters, one can achieve faster convergence to high-frequency geometric details, effectively leading to more nuanced surface reconstruction and a smaller reconstruction error.

**Reconstruction error.** We proceed by investigating the bandwidth's influence on the $L_2$ reconstruction error, which is defined (and can be bounded) as

$$\|f - \hat{f}\|_{L_2} = \left(\int_\Omega \left(f(x) - \hat{f}(x)\right)^2 \mathrm{d}x\right)^{1/2} \leq C_{\mathcal{X},\Omega}^{(\nu+d)/2} \|f\|_{H_\nu}, \tag{10}$$

where $C_{\mathcal{X},\Omega}$ is a constant that only depends on $\mathcal{X}$ and $\Omega$, and

$$\|f\|_{H_\nu}^2 = h^{2\nu} \left((2\pi)^{d/2} C_{d,\nu}\right)^{-1} \int_{\mathbb{R}^d} \left(\frac{2\nu}{h^2} + (2\pi\|\omega\|)^2\right)^{\nu+d/2} |\mathcal{F}[f](\omega)|^2 \mathrm{d}\omega \tag{11}$$

is the norm of the RKHS induced by a Matérn kernel with bandwidth $h$ and smoothness $\nu$. Moreover, $C_{d,\nu} := (2^d \pi^{d/2} \Gamma(\nu + d/2)(2\nu)^\nu)/\Gamma(\nu)$, and $\mathcal{F}[f]$ denotes the Fourier transform of a function $f$.

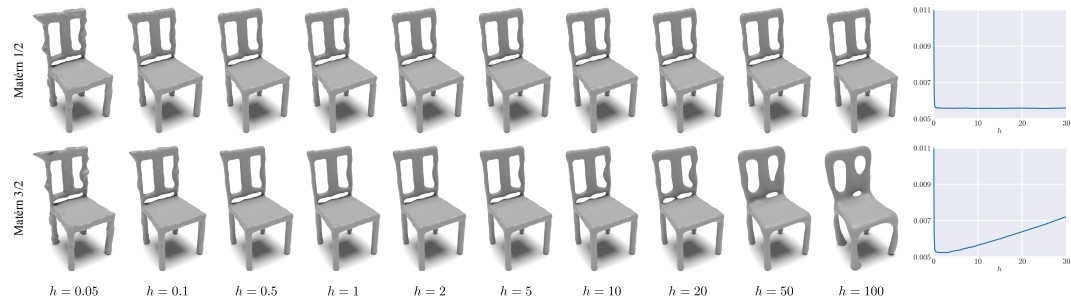

Figure 4: **Matérn kernels are tunable**. Surface reconstructions can be improved in practice by varying the bandwidth parameter, $h$, effectively tuning the kernels' EDR (see also Figure 3). However, setting $h$ too small ($< 0.5$) results in overfitting, while setting $h$ too big ($> 10$) oversmooths (*i.e.*, underfits) the true surface. This is also reflected in the reconstruction error (measured using Chamfer distance), plotted as a function of $h$ on the right.

For more information on the bound, including details on $C_{\mathcal{X},\Omega}$, please see, *e.g.*, Santin & Schaback (2016). By inspecting Eq. (10), we notice that the magnitude of the RKHS norm has a considerably high influence on the reconstruction error. Based on this observation and the fact that $\|f\|_{H_\nu}$ depends on $h$, our goal is now to further study the effect of $h$ on the RKHS norm. As a first result, we have:

**Proposition 2.** *The RKHS norm of Matérn kernels as defined in Eq. (11) can be bounded by*

$$\|f\|_{H_\nu}^2 \leq h^{2\nu} \left( \frac{1}{h^{2\nu+d}} C_{d,\nu}^1(\mathcal{F}[f]) + C_{d,\nu}^2(\mathcal{F}[f]) \right),$$

*where $h > 0$, and $C_{d,\nu}^1$ and $C_{d,\nu}^2$ are functions of $\mathcal{F}[f]$ that do not depend on $h$.*

A proof can be found in Appendix B. We immediately see that $\|f\|_{H_\nu}^2 \to \infty$ in either cases, $h \to 0$ and $h \to \infty$. Moreover, based on Proposition 2, it is easy to show that the norm, as a function of $h$, decreases as $h \to h^*$ from the left, and increases as $h \to \infty$, where $h^*$ is given as

$$h^* = \left( \frac{d}{2\nu} Q \right)^{1/(2\nu+d)} \quad \text{with} \quad Q = \frac{C_{d,\nu}^1(\mathcal{F}[f])}{C_{d,\nu}^2(\mathcal{F}[f])} \tag{12}$$

(see Appendix C for details). Our analysis shows that the reconstruction error can *not* be made arbitrarily small by just lowering $h$; if $h$ is chosen too small, the reconstruction error starts to increase again, *i.e.*, it *overfits* the true surface. Conversely, if $h$ is set too high, the resulting surface is too smooth (*underfits* the true surface), ultimately leading to increased errors. The optimal trade-off is reached if $h = h^*$. This is also observed in practice, see Figure 4. As seen from Figure 5, however, the described effect is more pronounced for smoother Matérn kernels (large $\nu$), as, at some point, the norm (hence, the reconstruction error) increases rapidly for large $h$. Our theoretical analysis reveals several practical insights. First, the reconstruction error is generally more sensitive to very small values of $h$. Second, a good starting point is always $h = 1$.

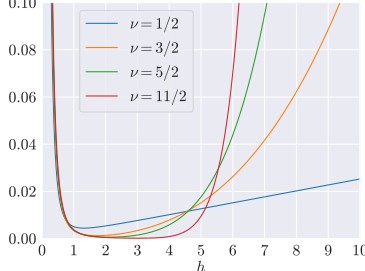

Figure 5: **RKHS norm as a function of $h$**. We plot the bound from Proposition 2 as a function of $h$.

### 3.3 RELATION TO NEURAL NETWORKS AND THE ARC-COSINE KERNEL

**Relation to neural networks.** Next, we study the connection between Matérn kernels and neural networks. It is well known that the first-order arc-cosine kernel mimics the computation of a two-layer, infinite-width ReLU network $f : \mathbb{R}^d \longrightarrow \mathbb{R}$, $f(x) = m^{-1/2} \sum_{i=1}^m v_i [w_i^\top x + b_i]_+$ when the bottom layer weights $W = (w_1, w_2, \ldots, w_m) \in \mathbb{R}^{d \times m}$ and biases $b = (b_1, b_2, \ldots, b_m) \in \mathbb{R}^m$ are *fixed* from initialization and drawn from a standard normal distribution, see, *e.g.*, Cho & Saul (2009); Williams et al. (2021). Here, $[x]_+ := \max\{0, x\}$ denotes the ReLU activation function.

Under similar assumptions, we now show that Matérn kernels mimic the computation of two-layer SIRENs (Sitzmann et al., 2020) if the layer-width approaches infinity. We claim the following:

**Theorem 3.** *Consider a two-layer fully-connected network* $f : \mathbb{R}^d \longrightarrow \mathbb{R}$ *with sine activation function,* $m$ *hidden neurons, and fixed bottom layer weights* $W = (w_1, w_2, \ldots, w_m) \in \mathbb{R}^{d \times m}$ *and biases* $b = (b_1, b_2, \ldots, b_m) \in \mathbb{R}^m$. *Let* $h, \nu > 0$ *and* $C_{d,\nu}$ *be defined as in Eq. (11). If* $w_i$ *is randomly initialized from*

$$p_\nu(\omega) = h^{-2\nu} C_{d,\nu} \left( \frac{2\nu}{h^2} + (2\pi \|\omega\|)^2 \right)^{-(\nu + d/2)}$$

*and* $b_i \sim \mathcal{U}(0, 2\pi)$, *the NTK of* $f$ *is a Matérn kernel with bandwidth* $h$ *and smoothness* $\nu$ *as* $m \to \infty$.

A proof can be found in Appendix D. This result establishes for the first time a connection between SIRENs and kernel methods. While explicitly shown for Matérn kernels, as we argue in the Appendix D, the arguments of Theorem 3 in fact apply to all stationary kernels, making it a powerful tool to study the connection between widely used SIRENs and any stationary kernel function.

**Matérn kernels vs. arc-cosine kernels.** Lastly, we aim to compare Matérn kernels with the previously used first-order arc-cosine kernel. Our analysis is based on:

**Theorem 4** (Chen & Xu (2021), Theorem 1). *When restricted to the hypersphere* $\mathbb{S}^{d-1}$, *the RKHS of the Matérn kernel with smoothness* $\nu = 1/2$ *(the Laplace kernel) include the same set of functions as the RKHS induced by the NTK of a fully-connected* $(L \geq 2)$*-layer ReLU network.*

In other words, Theorem 4 shows that the RKHS of the Laplace kernel (a Matérn kernel with smoothness $\nu = 1/2$) is the same as the RKHS induced by the NTK of a fully-connected ReLU network when inputs are restricted to $\mathbb{S}^{d-1}$. Based on Theorem 4, we obtain the following connection between Laplace and the first-order arc-cosine kernel:

**Corollary 5.** *When restricted to the hypersphere* $\mathbb{S}^{d-1}$, *(1) the RKHS of the Matérn kernel with smoothness* $\nu = 1/2$ *(the Laplace kernel) is equal to the RKHS induced by the first-order arc-cosine kernel, implying that (2) the Laplace and arc-cosine kernel are equal up to an affine transformation.*

See Appendix E for a proof. For general input domains, our empirical results presented next support (at least for $(L = 2)$-layer networks) the widely believed claim that the NTK is not significantly different from standard kernels (Belkin et al., 2018; Hui et al., 2019; Geifman et al., 2020), such as the Laplace kernel. Indeed, we show that the Laplace kernel even outperforms the arc-cosine kernel.

## 4 EXPERIMENTS AND RESULTS

We systematically evaluated the effectiveness of Matérn kernels in the context of implicit surface reconstruction, presenting results on ShapeNet (Chang et al., 2015) and the *Surface Reconstruction Benchmark* (SRB; Berger et al. (2013)) in Section 4.1. Moreover, Section 4.2 demonstrates Matérn kernels' ability to reconstruct high-frequency textures on the *Google Scanned Objects* (GSO; Downs et al. (2022)) dataset and Objaverse (Deitke et al., 2023). Leveraging Neural Kernel Fields (Williams et al., 2022), we present and evaluate learnable Matérn kernels in Section 4.3.

### 4.1 SURFACE RECONSTRUCTION ON SHAPENET AND SRB

**ShapeNet**. We compare Matérn kernels in a sparse setting against the arc-cosine kernel on ShapeNet (using train/val/test split provided by Mescheder et al. (2019)). To do so, we randomly sample $m = 1,000$ surface points with corresponding normals for each shape. We implemented Matérn kernels in PyTorch and took the official CUDA implementation of the arc-cosine kernel from NS, eventually integrated into a *unified* framework to ensure fair comparison. Notably, we did not use Nyström sampling as in NS and employed PyTorch's direct Cholesky solver to solve the KRR problem in Eq. (1) instead of an iterative conjugate gradient solver. Moreover, similar to NS, we approximate the gradient part in Eq. (1) with finite differences. See Appendix F for details.

Quantitative results in terms of F-Score (with a threshold of 0.01) and Chamfer distance (CD; always reported $\times 10^3$) can be found in Table 1. Matérn 1/2 and 3/2 consistently outperform the arc-cosine kernel as well as popular SPSR (Kazhdan & Hoppe, 2013). In addition, our experiments show

| | F-Score ↑ | | | CD ↓ | | |
|---|---|---|---|---|---|---|
| | 0.5 | 1 | 2 | 0.5 | 1 | 2 |
| Matérn 1/2 | 93.6 | 93.7 | 93.7 | 4.05 | 4.02 | 4.02 |
| Matérn 3/2 | 94.6 | 94.8 | **94.9** | 4.05 | **4.00** | 4.09 |
| Matérn 5/2 | 92.4 | 93.8 | 92.9 | 6.42 | 5.65 | 6.91 |
| Matérn ∞ | 59.3 | 52.4 | 40.9 | 57.56 | 50.59 | 48.38 |
| Arc-cosine | | 92.8 | | | 4.67 | |
| NS | | 90.6 | | | 4.74 | |
| SPSR | | 84.3 | | | 6.26 | |

| | CD ↓ | HD ↓ |
|---|---|---|
| Matérn 1/2 | 0.21 | 4.43 |
| Matérn 3/2 | 0.18 | 2.93 |
| Matérn 5/2 | 0.58 | 22.52 |
| Matérn ∞ | 3.83 | 33.27 |
| NS* | 0.19 | 3.19 |
| NS | **0.17** | **2.85** |
| SPSR | 0.21 | 4.69 |
| FFN | 0.28 | 4.45 |
| SIREN | 0.19 | 3.86 |
| SAP | 0.21 | 4.85 |
| DiGS | 0.18 | 3.55 |
| VisCo | 0.18 | 2.95 |
| OG-INR | 0.20 | 4.06 |

Table 1: **Results on ShapeNet (left) and SRB (right) for non-learnable kernels**. Arc-cosine kernel and Matérn $\nu \in \{1/2, 3/2, 5/2, \infty\}$ are implemented in a unified framework and based on the exact same parameters. "NS*" denotes the best result we could achieve by running the official implementation of NS, see Appendix G. **Bold** marks best result, underline second best.

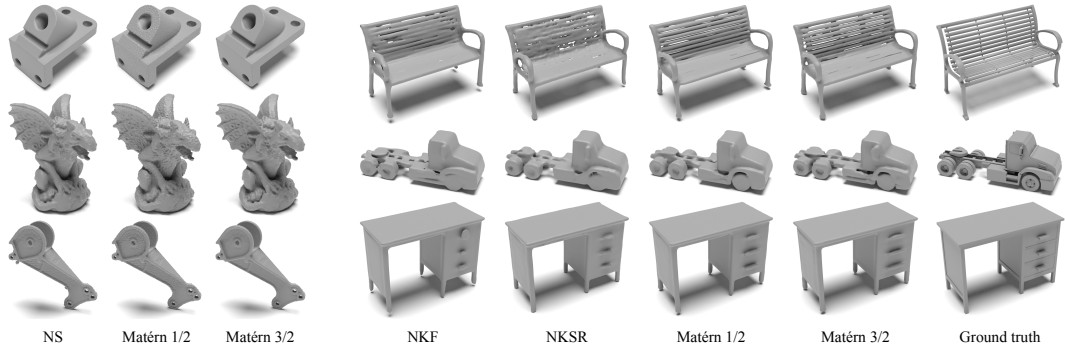

| NS | Matérn 1/2 | Matérn 3/2 | NKF | NKSR | Matérn 1/2 | Matérn 3/2 | Ground truth |

Figure 6: **Qualitative results for non-learnable kernels on SRB (left) and learned kernels on ShapeNet (right)**. Please see Appendices G and I for more qualitative results.

that the bandwidth parameter, $h$, can be used to tune the surface reconstruction (lower the error), while not being too sensitive in practice. Matérn 5/2 and the Gaussian kernel (for $\nu \to \infty$) perform significantly worse, being simply too smooth. Figure 2 and Appendix F shows qualitative results.

**SRB**. Next, we evaluate Matérn kernels on the challenging Surface Reconstruction Benchmark which consists of five complex shapes simulated from incomplete and noisy range scans with up to 100,000 points. For this, we implemented a highly scalable version of Matérn kernels, closely following the NS framework to solve the KRR problem in Eq. (1) using Nyström sampling (with 15,000 samples) and the FALKON (Rudi et al., 2017) solver. We compare against NS and SPSR, as well as kernel-free methods, Fourier Feature Networks (FFNs; Tancik et al. (2020)), SIREN, SAP (Peng et al., 2021), DiGS (Ben-Shabat et al., 2022), VisCo (Pumarola et al., 2022), and OG-INR (Koneputugodage et al., 2023). All methods are optimization-based (FFN and SIREN are overfitted to a single shape) and use surface normals. For Matérn kernels, we did a modest parameter sweep over $h \in \{0.5, 1, 2\}$ and took the reconstruction with the lowest Chamfer distance. Runtime is measured on a single NVIDIA V100.

| | Time |
|---|---|
| Matérn 1/2 | 9.85 |
| Matérn 3/2 | 12.47 |
| Matérn 5/2 | 15.26 |
| Matérn ∞ | 12.76 |
| NS* | 19.89 |
| NS | 11.91 |
| SPSR | **1.65** |

Table 2: **Runtime comparison on SRB**. Time in seconds.

Quantitative results obtained by employing Chamfer and Hausdorff distance (HD) can be found in Table 1, demonstrating that Matérn 3/2 performs on par with NS while being significantly faster to compute (compared to what *we* measured for NS), see Table 2. Moreover, Matérn 3/2 outperforms all evaluated state-of-the-art kernel-free methods (FFN, SIREN, SAP, DiGS, VisCo, and OG-INR). Figure 6 (left) and Appendix G provide more qualitative results and per-object metrics.

## 4.2 Texture Reconstruction on GSO and Objaverse

Lastly, we demonstrate Matérn kernels' ability to represent other *high-frequency* scene attributes, such as texture. To do so, we randomly sample $m$ surface points with corresponding normals and per-point RGB color values from textured meshes, yielding an extended dataset $\mathcal{D}' = \mathcal{D} \times \{c_1, c_2, \ldots, c_m\} \subset \Omega \times \mathbb{R}^d \times \mathbb{R}^3$ for each shape. Then, instead of modeling an SDF $f$ as in Eq. (1), we are seeking a function $f' : \mathbb{R}^d \longrightarrow \mathbb{R}^4$ that, along with signed distances, also predicts per-point RGB values. Please find more information in Appendix H about how we adapt

| $m$ | Kernel | PSNR ↑ | LPIPS ↓ |
|---|---|---|---|
| 10,000 | Matérn 3/2 | **19.61** | **2.05** |
| | Arc-cosine | 19.34 | 2.07 |
| 2,500 | Matérn 3/2 | **18.92** | 2.15 |
| | Arc-cosine | 18.88 | **2.09** |

Table 3: **Texture reconstruction on GSO**. Matérn kernels outperform the arc-cosine kernel in dense and sparse settings.

the KRR problem in Eq. (1) to estimate $\hat{f}'$. Finally, we extract the object's surface using Marching Cubes and trilinearily interpolate RGB values at previously predicted surface points. We present quantitative results in terms of PSNR and LPIPS on GSO in Table 3 (metrics evaluated on the texture atlas; see Appendix H for further details). Matérn 3/2 surpasses the arc-cosine kernel in the densely sampled setting using 10,000 surface points, as well as in the sparse setting. Notably, we did not tune Matérn kernels' bandwidth parameter for this experiment.

We also show qualitative results on the challenging Objaverse (Deitke et al., 2023) dataset in Figure 7. We chose Objaverse as it includes extremely high-resolution and complex textures. Matérn 3/2 reconstructs high-frequency texture details with great precision, overall yielding visually more pleasant reconstructions than the arc-cosine kernel. Reconstructed textures have fewer artifacts and are generally sharper, demonstrating that Matérn kernels can overcome spectral bias much better than the arc-cosine kernel. Note that this perfectly confirms our theoretical analysis presented in Section 3.2.

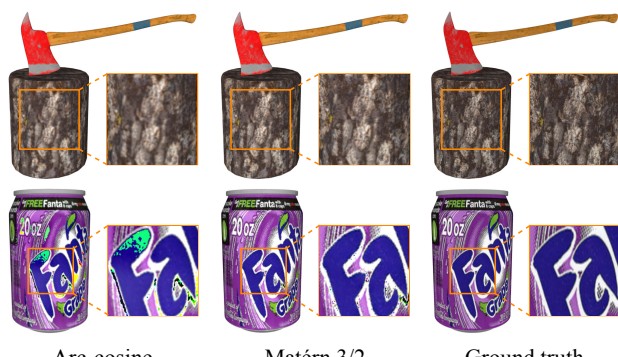

Arc-cosine      Matérn 3/2      Ground truth

Figure 7: **Texture reconstruction on Objaverse**. Matérn kernels lead to fewer artifacts and sharper details than the arc-cosine kernel.

## 4.3 Data-dependent Matérn Kernels

We leverage the *Neural Kernel Field* (NKF) framework introduced by Williams et al. (2022) to make Matérn kernels learnable. Specifically, NKF proposed to feed points $x, y \in \mathbb{R}^d$ through an input-conditioned neural network $\varphi : \mathbb{R}^d \longrightarrow \mathbb{R}^q$ before evaluating a kernel function, $k^\varphi(x, y; \mathcal{D}) = k([x, \varphi(x; \mathcal{D})], [y, \varphi(y; \mathcal{D})])$. We re-implemented the NKF framework since the authors did not provide code. We set $q = 32$ for our experiments. Please see Williams et al. (2022) for details.

**Sparse surface reconstruction and extreme generalization.** We compare learned Matérn kernels against the original NKF framework (with the arc-cosine kernel) and NKSR (Huang et al., 2023) in a sparse setting on ShapeNet. Again, we sample $m = 1,000$ surface points and corresponding normals for each shape and set $h = 1$ for all Matérn kernels. Additionally, we evaluate Matérn kernels' out-of-category generalization ability in an extreme setting, in which we train only on chairs and evaluate on the other 12 ShapeNet categories. Table 4 reports the results, quantified using intersection-over-union (IoU), Chamfer distance, and normal consistency (NC). Learned Matérn 1/2 and 3/2 outperform the arc-cosine kernel while being significantly faster to train. Training NKF on the entire ShapeNet dataset (consisting of approx. 30,000 shapes) takes about six days—almost twice as long as for Matérn kernels, which require about three days (measured on a single NVIDIA A100). Additionally, although Matérn 1/2 is not able to surpass NKSR, it comes very close (95.3 vs. 95.6 in IoU) while having a shorter training time. Regarding out-of-category generalization, we observe that Matérn 1/2 performs best, followed by Matérn 3/2, NKF, and NKSR. Please see Figure 6 (right) and Appendix I for qualitative results.

| | IoU ↑ | CD ↓ | NC ↑ | **Train only on chairs, test on all** | | | |
|---|---|---|---|---|---|---|---|
| | | | | IoU ↑ | CD ↓ | NC ↑ | Time/epoch |
| Matérn 1/2 | 95.3 | 2.58 | **95.6** | **93.4** | 3.06 | **94.3** | **7.71 min** |
| Matérn 3/2 | 94.9 | 2.70 | 95.3 | 92.8 | 3.34 | 94.2 | 8.26 min |
| Matérn 5/2 | 93.3 | 3.07 | 94.9 | 90.5 | 4.11 | 93.9 | 8.56 min |
| Matérn ∞ | 92.1 | 3.39 | 94.2 | 84.7 | 6.70 | 92.6 | 7.74 min |
| NKF* | 94.7 | 2.70 | 95.2 | 92.8 | 3.30 | 94.1 | 31.98 min |
| NKF | 94.7 | 2.65 | 95.4 | | | | |
| NKSR | **95.6** | **2.34** | 95.4 | 89.6 | **2.70** | 94.1 | 41.66 min |

Table 4: **Results on ShapeNet for learned kernels**. "NKF*" denotes a re-implemented variant of NKF. NKF* and Matérn $\nu \in \{1/2, 3/2, 5/2, \infty\}$ are based on the same framework and parameters; they differ *only* in the employed kernel. Runtime is measured on a single NVIDIA A100 with a batch size of one to ensure fair comparison. **Bold** marks best result, underline second best.

| | No noise ($\sigma = 0$) | | | | | | Small noise ($\sigma = 0.0025$) | | | | | | Big noise ($\sigma = 0.005$) | | | | | |
|---|---|---|---|---|---|---|---|---|---|---|---|---|---|---|---|---|---|---|
| | IoU ↑ | | | CD ↓ | | | IoU ↑ | | | CD ↓ | | | IoU ↑ | | | CD ↓ | | |
| | 0.5 | 1 | 2 | 0.5 | 1 | 2 | 0.5 | 1 | 2 | 0.5 | 1 | 2 | 0.5 | 1 | 2 | 0.5 | 1 | 2 |
| Matérn 1/2 | 93.3 | **93.6** | 93.5 | 2.88 | 2.85 | 2.87 | 91.9 | **92.1** | 92.1 | 3.03 | 3.04 | 3.07 | 89.5 | **89.6** | 89.6 | 3.37 | **3.36** | 3.39 |
| Matérn 3/2 | 92.7 | 92.7 | 92.3 | 3.08 | 3.20 | 3.25 | 89.7 | 89.8 | 89.7 | 3.48 | 3.69 | 3.54 | 85.8 | 85.5 | 86.3 | 4.23 | 4.32 | 4.15 |
| Matérn 5/2 | 91.8 | 90.5 | 88.8 | 3.58 | 3.69 | 4.18 | 87.9 | 87.6 | 87.0 | 4.29 | 4.48 | 4.43 | 83.4 | 82.2 | 84.3 | 4.90 | 5.03 | 5.11 |
| Matérn ∞ | 86.4 | 85.8 | 86.8 | 5.80 | 5.07 | 4.75 | 81.3 | 83.3 | 81.0 | 7.27 | 5.62 | 6.45 | 76.7 | 80.7 | 79.3 | 8.27 | 5.90 | 6.67 |
| NKF* | | 93.2 | | | 2.97 | | | 92.0 | | | 3.12 | | | 89.5 | | | 3.39 | |
| NKSR | | 91.1 | | | **2.65** | | | 90.1 | | | **2.98** | | | 88.4 | | | 3.41 | |

Table 5: **Robustness against noise**. We compare Matérn kernels' robustness against noise on a subset of the ShapeNet dataset using different noise levels, $\sigma \in \{0, 0.0025, 0.005\}$ and bandwidths, $h \in \{0.5, 1, 2\}$. **Bold** marks best result, underline second best.

**Robustness against noise.** We evaluate Matérn kernel's robustness against different noise levels, $\sigma \in \{0, 0.0025, 0.005\}$, on a subset of the ShapeNet dataset which includes approximately 1,700 shapes. To construct the dataset, we downsampled each ShapeNet category to include only 5% of the shapes. Table 5 presents the results. NKSR, being specifically optimized to deal with noisy inputs, degrades the least with increasing noise level (3% in IoU from no to big noise versus 4.3% for Matérn 1/2). Moreover, Matérn 1/2 is slightly more robust against noise than NKF. Generally, varying the bandwidth, $h$, helps increase the robustness against noise.

**Convergence speed.** Finally, we analyze Matérn kernels' convergence speed in comparison to the arc-cosine kernel as implemented in NKF, see inset figure on the right. As exemplified, Matérn 1/2 already converges after just 100 epochs, while the arc-cosine kernel requires twice as many epochs, also never being able to reach the same top accuracy. In general, we see that the smoother the kernel, the slower the convergence—an observation that perfectly matches our theoretical analysis in Section 3.2. Matérn 1/2 converges the fastest, followed by Matérn 3/2, 5/2, and the Gaussian kernel.

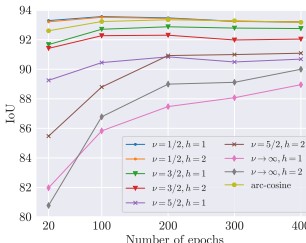

## 5 CONCLUSION

In this work, we have proposed to use the family of Matérn kernels for implicit surface reconstruction and showed that it consistently outperforms the recently employed first-order arc-cosine kernel—both, in a non-learnable as well as learnable regime—while being significantly easier to implement, faster to compute and train, and highly scalable. We demonstrated that Matérn kernels lead to *tunable* surface reconstruction, and, based on a new bound of the $L_2$ reconstruction error, derive practical insights into how to choose their tunable bandwidth parameter. Moreover, we presented an in-depth theoretical analysis, analyzing Matérn kernels' relation to widely used Fourier feature mappings, SIREN networks, and arc-cosine kernels.

ACKNOWLEDGMENTS

We thank Francis Williams for valuable discussions and his support in re-implementing the NKF framework and Lukas Meyer for his help during NKSR evaluation. This work was funded by the German Federal Ministry of Education and Research (BMBF), FKZ: 01IS22082 (IRRW). The authors are responsible for the content of this publication. The authors gratefully acknowledge the scientific support and HPC resources provided by the Erlangen National High Performance Computing Center (NHR@FAU) of the Friedrich-Alexander-Universität Erlangen-Nürnberg (FAU) under the NHR project b112dc IRRW. NHR funding is provided by federal and Bavarian state authorities. NHR@FAU hardware is partially funded by the German Research Foundation (DFG) – 440719683.

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

## A  ON THE STATIONARITY OF MATÉRN KERNELS

As noted in Section 3.1 of the main paper, Matérn kernels are *stationary*: they only depend on the difference $x - y$ between two points $x, y \in \Omega$, hence being translation invariant. This is because $k(x - y) = k(x + t - (y + t))$ for $t \in \mathbb{R}^d$ and all stationary kernels, $k$. In contrast, the arc-cosine kernel is not translation invariant as its value depends on the absolute positions, $x, y$, see Eq. (3) of the main paper (*i.e.*, the arc-cosine kernel can not be written as a function of $x - y$). In fact, the arc-cosine kernel is *non-stationary*. In the following and with the help of Figure 8, we will further discuss why stationary kernels, such as Matérn kernels, might be a better choice for kernel-based implicit surface reconstruction than non-stationary kernels, such as the arc-cosine kernel.

First, stationary kernels are *naturally* translation invariant. Consequently, they are independent of the absolute embedding of the input points—translating them does not change the reconstructed surface as the kernel's value only depends on the *relative* distances between points. This is highly beneficial from a practical perspective since no additional steps (such as centering the point cloud) have to be taken before surface reconstruction. Being non-stationary, the arc-cosine kernel behaves differently. If we would just use the kernel as in Eq. (3) of the main paper, the reconstructed surface would vary depending on the actual location of the input points in space. This is a highly undesirable property when it comes to surface reconstruction. To make the arc-cosine kernel translation invariant, input points must be centered before reconstruction (thus eliminating the effects of translation). A centered arc-cosine kernel, denoted as $\bar{k}_{\text{AC}}$, reads:

$$\bar{k}_{\text{AC}}(x, y) = \frac{\|x - \bar{c}\|\|y - \bar{c}\|}{\pi} \left( \sin \bar{\theta} + (\pi - \bar{\theta}) \cos \bar{\theta} \right), \quad \text{where} \quad \bar{\theta} = \cos^{-1} \left( \frac{(x - \bar{c})^\top (y - \bar{c})}{\|x - \bar{c}\|\|y - \bar{c}\|} \right).$$

Here, $c \in \mathbb{R}^d$ denotes the centroid of the input point cloud. $\bar{k}_{\text{AC}}$ is obviously translation invariant because $\|x + t - (\bar{c} + t)\|\|y + t - (\bar{c} + t)\| = \|x - \bar{c}\|\|y - \bar{c}\|$ and $(x + t - (\bar{c} + t))^\top (y + t - (\bar{c} + t)) = (x - \bar{c})^\top (y - \bar{c})$ for all $t \in \mathbb{R}^d$. This follows because translating the input points also shifts its centroid. In conclusion, while stationary Matérn kernels can be directly used for kernel-based surface reconstruction, additional pre-processing steps are necessary for the arc-cosine kernel.

Second, stationary kernels lead to *locally consistent* surface reconstructions. Since a stationary kernel's value only depends on the *relative* positions, objects in a scene (or parts of objects) with similar geometric properties will be reconstructed consistently as long as the relative distances between points on each instance (or part) remain the same. This is a crucial and very desirable property for surface reconstruction which is *not* shared by the arc-cosine kernel (because it is non-stationary).

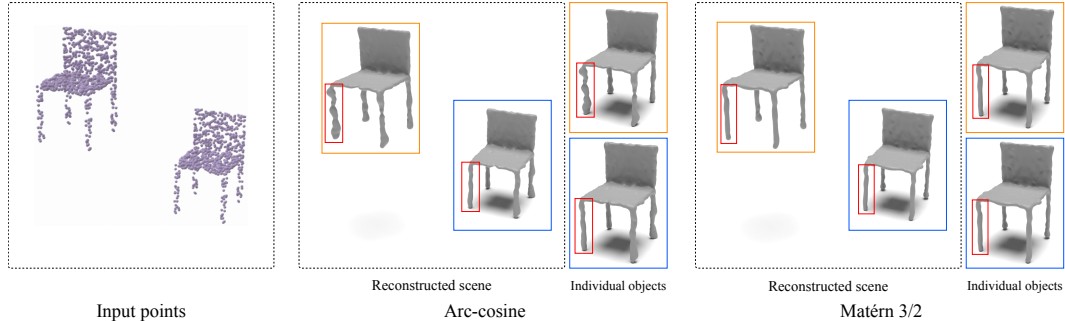

Figure 8: **Matérn kernels are translation invariant and lead to locally consistent surface reconstructions**. Here, we show that two identical chairs in a scene are reconstructed differently depending on the absolute position of the input points when using the non-stationary arc-cosine kernel (especially noticeable at the chair's legs). In contrast, stationary Matérn kernels lead to locally consistent surface reconstructions, being independent of the actual embedding (*i.e.*, both chairs have the exact same reconstructed surface).

## B    PROOF OF PROPOSITION 2

**Proposition 2.** *The RKHS norm of Matérn kernels as defined in Eq. (11) can be bounded by*

$$\|f\|^2_{H_\nu} \le h^{2\nu}\left(\frac{1}{h^{2\nu+d}}C^1_{d,\nu}(\mathcal{F}[f]) + C^2_{d,\nu}(\mathcal{F}[f])\right),$$

*where $h > 0$, and $C^1_{d,\nu}$ and $C^2_{d,\nu}$ are functions of $\mathcal{F}[f]$ that do not depend on $h$.*

*Proof.*  Based on a simple bound

$$
\begin{aligned}
I &= \int_{\mathbb{R}^d}\left(\frac{2\nu}{h^2} + (2\pi\|\omega\|)^2\right)^{\nu+d/2}|\mathcal{F}[f](\omega)|^2\mathrm{d}\omega\\
&\le \int_{\mathbb{R}^d}\left(\frac{2\nu}{h^2}\right)^{\nu+d/2}|\mathcal{F}[f](\omega)|^2\mathrm{d}\omega + \int_{\mathbb{R}^d}\left((2\pi\|\omega\|)^2\right)^{\nu+d/2}|\mathcal{F}[f](\omega)|^2\mathrm{d}\omega\\
&= \left(\frac{2\nu}{h^2}\right)^{\nu+d/2}\underbrace{\int_{\mathbb{R}^d}|\mathcal{F}[f](\omega)|^2\mathrm{d}\omega}_{=:C'(\mathcal{F}[f])} + \underbrace{\int_{\mathbb{R}^d}\left((2\pi\|\omega\|)^2\right)^{\nu+d/2}|\mathcal{F}[f](\omega)|^2\mathrm{d}\omega}_{=:C'_{d,\nu}(\mathcal{F}[f])}\\
&= \left(\frac{2\nu}{h^2}\right)^{\nu+d/2}C'(\mathcal{F}[f]) + C'_{d,\nu}(\mathcal{F}[f])
\end{aligned}
$$

we get

$$
\begin{aligned}
\|f\|^2_{H_\nu} &= h^{2\nu}\left((2\pi)^{d/2}C_{d,\nu}\right)^{-1}I\\
&\le h^{2\nu}\left((2\pi)^{d/2}C_{d,\nu}\right)^{-1}\left(\left(\frac{2\nu}{h^2}\right)^{\nu+d/2}C'(\mathcal{F}[f]) + C'_{d,\nu}(\mathcal{F}[f])\right)\\
&= h^{2\nu}\left(\frac{1}{h^{2\nu+d}}\underbrace{(2\nu)^{\nu+d/2}\left((2\pi)^{d/2}C_{d,\nu}\right)^{-1}C'(\mathcal{F}[f])}_{=:C^1_{d,\nu}(\mathcal{F}[f])} + \underbrace{\left((2\pi)^{d/2}C_{d,\nu}\right)^{-1}C'_{d,\nu}(\mathcal{F}[f])}_{=:C^2_{d,\nu}(\mathcal{F}[f])}\right)\\
&= h^{2\nu}\left(\frac{1}{h^{2\nu+d}}C^1_{d,\nu}(\mathcal{F}[f]) + C^2_{d,\nu}(\mathcal{F}[f])\right)
\end{aligned}
$$

which concludes the proof.    □

## C    DERIVATION OF $h^*$ IN EQ. (12)

Taking derivative of the bound presented in Proposition 2 w.r.t. $h$ yields

$$\frac{\mathrm{d}}{\mathrm{d}h}\left(h^{2\nu}\left(\frac{1}{h^{2\nu+d}}C^1_{d,\nu}(\mathcal{F}[f]) + C^2_{d,\nu}(\mathcal{F}[f])\right)\right) = \frac{1}{h^{d+1}}\left(2\nu C^2_{d,\nu}(\mathcal{F}[f])h^{2\nu+d} - dC^1_{d,\nu}(\mathcal{F}[f])\right).$$

Since $d, h > 0$, the leading factor $1/h^{d+1}$ never becomes zero, so we must have

$$2\nu C^2_{d,\nu}(\mathcal{F}[f])h^{2\nu+d} - dC^1_{d,\nu}(\mathcal{F}[f]) = 0 \iff h^* = \left(\frac{d}{2\nu}\frac{C^1_{d,\nu}(\mathcal{F}[f])}{C^2_{d,\nu}(\mathcal{F}[f])}\right)^{1/(2\nu+d)}$$

which shows what we have stated in Eq. (12) of the main paper.

## D    PROOF OF THEOREM 3

The proof of Theorem 3 is based on Bochner's theorem (in harmonic analysis) which reads:

**Theorem 6** (Bochner). *A continuous function $k : \mathbb{R}^d \longrightarrow \mathbb{R}$ with $k(0) = 1$ is positive definite if and only if there exists a finite positive Borel measure $\mu$ on $\mathbb{R}^d$ such that*

$$k(\tau) = \int_{\mathbb{R}^d} e^{i\omega^\top \tau} \mathrm{d}\mu(\omega).$$

In essence, Bochner's theorem states that $k$ and $\mu$ are *Fourier duals*; given a stationary kernel $k$, one can obtain $\mu$ (better known as *spectral density* if normalized) by applying the *inverse* Fourier transform to $k$. On the other hand, given a spectral density $\mu$, one can obtain the corresponding kernel function $k$ by applying the Fourier transform to $\mu$. Based on Bochner's theorem,

$$k(\tau) = k(x - y) = \int_{\mathbb{R}^d} e^{i\omega^\top (x-y)} \mathrm{d}\mu(\omega) = \mathbb{E}_{\omega \sim \mu}\left[\cos(\omega^\top (x - y))\right]$$

because $k$ is real and for *even* spectral densities. On the other hand, Rahimi & Recht (2007) showed

$$
\begin{aligned}
k(\tau) = k(x - y) &= \mathbb{E}_{\omega \sim \mu}[\cos(\omega^\top (x - y))] \\
&= \mathbb{E}_{\omega \sim \mu}[\cos(\omega^\top (x - y))] + \underbrace{\mathbb{E}_{\omega \sim \mu, b \sim \mathcal{U}(0,2\pi)}[\cos(\omega^\top (x + y) + 2b)]}_{= 0\,(*)} \\
&= \mathbb{E}_{\omega \sim \mu, b \sim \mathcal{U}(0,2\pi)}[\cos(\omega^\top (x - y)) + \cos(\omega^\top (x + y) + 2b)] \\
&= \mathbb{E}_{\omega \sim \mu, b \sim \mathcal{U}(0,2\pi)}[2\cos(\omega^\top x + b)\cos(\omega^\top y + b)],
\end{aligned}
$$

where $(*)$ follows because $b$ is uniformly distributed in $[0, 2\pi]$ (hence, the inner expectation w.r.t. $b$ becomes zero).

We will now prove Theorem 3, copied below for convenience:

**Theorem 3.** *Consider a two-layer fully-connected network $f : \mathbb{R}^d \longrightarrow \mathbb{R}$ with sine activation function, $m$ hidden neurons, and fixed bottom layer weights $W = (w_1, w_2, \ldots, w_m) \in \mathbb{R}^{d \times m}$ and biases $b = (b_1, b_2, \ldots, b_m) \in \mathbb{R}^m$. Let $h, \nu > 0$ and $C_{d,\nu}$ be defined as in Eq. (11). If $w_i$ is randomly initialized from*

$$p_\nu(\omega) = h^{-2\nu} C_{d,\nu} \left(\frac{2\nu}{h^2} + (2\pi\|\omega\|)^2\right)^{-(\nu+d/2)}$$

*and $b_i \sim \mathcal{U}(0, 2\pi)$, the NTK of $f$ is a Matérn kernel with bandwidth $h$ and smoothness $\nu$ as $m \to \infty$.*

*Proof.* Let $f : \mathbb{R}^d \longrightarrow \mathbb{R}$ be a two-layer fully-connected network with sine activation function, *i.e.*,

$$f(x) = \sqrt{\frac{2}{m}} \sum_{i=1}^{m} v_i \sin(w_i^\top x + b_i)$$

with bottom-layer weights $W = (w_1, w_2, \ldots, w_m) \in \mathbb{R}^{d \times m}$, biases $b = (b_1, b_2, \ldots, b_m) \in \mathbb{R}^m$, and top-layer weights $v = (v_1, v_2, \ldots, v_m) \in \mathbb{R}^m$. Assume that $W$ and $b$ are *fixed* (that is, we only allow the second layer to be trained), and initialized according to some distribution. Choose

$$w_i \sim p_\nu(\omega) = h^{-2\nu} C_{d,\nu} \left(\frac{2\nu}{h^2} + (2\pi\|\omega\|)^2\right)^{-(\nu+d/2)} \tag{13}$$

for $h, \nu > 0$ and $b_i \sim \mathcal{U}(0, 2\pi)$ for all $i \in \{1, 2, \ldots, n\}$. Then, based on the fact that $\partial_{v_i} f(x) = \sqrt{2}\sin(\omega_i^\top x + b_i)/\sqrt{m}$, it is easy to see that the NTK of $f$ is given by

$$
\begin{aligned}
k_{\mathrm{NTK}}(x, y) &= \sum_{i=1}^{m} \partial_{v_i} f(x) \partial_{v_i} f(y) \\
&= \frac{1}{m} \sum_{i=1}^{m} 2\sin(w_i^\top x + b_i)\sin(w_i^\top y + b_i) \\
&= \frac{1}{m} \sum_{i=1}^{m} 2\cos(w_i^\top x + b_i)\cos(w_i^\top y + b_i) - 2\cos(w_i^\top (x + y) + 2b_i) \\
&= \frac{1}{m} \sum_{i=1}^{m} 2\cos(w_i^\top x + b_i)\cos(w_i^\top y + b_i) - \frac{1}{m} \sum_{i=1}^{m} 2\cos(w_i^\top (x + y) + 2b_i).
\end{aligned}
$$

As $m \to \infty$, we finally obtain

$$k_{\text{NTK}}(x, y) = \mathbb{E}_{(w,b)}\left[2\cos(w^\top x + b)\cos(w^\top y + b)\right] - 2\underbrace{\mathbb{E}_{(w,b)}\left[\cos(w^\top(x+y) + 2b)\right]}_{=0 \; (*)}$$

$$\overset{(**)}{=} k_\nu(x, y).$$

Here, $(*)$ follows because $b$ is uniformly distributed, and $(**)$ since $w$ is distributed according to $p_\nu$ which is, in fact, the *spectral density* of $k_\nu$, see, *e.g.*, Eq. (10) in Kanagawa et al. (2018). $\qquad \square$

In fact, Theorem 3 can be generalized in that the NTK of a SIREN is generally any stationary kernel:

**Remark 7.** *The NTK of a* SIREN *can be associated with any stationary kernel if the bottom-layer weights in Eq. (13) are initialized according to the corresponding kernel's spectral density.*

This is a direct consequence of Bochner's theorem.

## E    PROOF OF COROLLARY 5

We first re-state the following result, needed to prove the second part of Corollary 5:

**Theorem 8** (Saitoh & Sawano (2016), Theorem 2.17). *Let $k_1$ and $k_2$ be two positive definite kernels. Denote by $H_1, H_2$ the RKHSs induced by $k_1$ and $k_2$. Then,*

$$H_1 \subset H_2 \iff \gamma^2 k_2 - k_1 \text{ is positive definite for } \gamma > 0.$$

We are now ready to prove Corollary 5 which is copied below for convenience:

**Corollary 5.** *When restricted to the hypersphere $\mathbb{S}^{d-1}$, (1) the RKHS of the Matérn kernel with smoothness $\nu = 1/2$ (the Laplace kernel) is equal to the RKHS induced by the first-order arc-cosine kernel, implying that (2) the Laplace and arc-cosine kernel are equal up to an affine transformation.*

*Proof.* (1) Follows from Theorem 4 by noting that the NTK of a fully-connected ($L = 2$)-layer ReLU network is the first-order arc-cosine kernel, see, *e.g.*, Cho & Saul (2009); Williams et al. (2021). (2) follows from (1) by noting that $H_{1/2} \subset H_{\text{AC}}$ when restricted to $\mathbb{S}^{d-1}$, so, based on Theorem 8, $k := \gamma^2 k_{\text{AC}} - k_{1/2}$ is a valid kernel for $\gamma > 0$. Re-arranging the equation, we have

$$k_{\text{AC}} = a k_{1/2} + b, \quad \text{where} \quad a = 1/\gamma^2, b = k/\gamma^2.$$

This shows that the Laplace and arc-cosine kernel are indeed related by an affine transformation when restricted to $\mathbb{S}^{d-1}$, concluding the proof. $\qquad \square$

## F    ADDITIONAL DETAILS FOR SHAPENET EXPERIMENT

In this section, we provide implementation details and additional qualitative results on the ShapeNet experiment as presented in Section 4.1 of the main paper.

### F.1    IMPLEMENTATION DETAILS

Following common practice (see, *e.g.*, Williams et al. (2022)), we use *finite differences* to approximate the gradient part of the KRR problem in Eq. (1) of the main paper. Specifically, denote $\mathcal{X}' := \mathcal{X}^+ \cup \mathcal{X}^-$, where $\mathcal{X}^+ := \{x_1 + \epsilon n_1, x_2 + \epsilon n_2, \dots, x_m + \epsilon n_m\}$ and $\mathcal{X}^- := \{x_1 - \epsilon n_1, x_2 - \epsilon n_2, \dots, x_m - \epsilon n_m\}$ for a fixed $\epsilon > 0$. The Representer Theorem (Kimeldorf & Wahba, 1970; Schölkopf et al., 2001) tells us that the solution to Eq. (1) is of the form

$$f(x) = \sum_{i=1}^{2m} \alpha_i k(x, x_i')$$

which is linear in the coefficients $\alpha \in \mathbb{R}^{2m}$, given by

$$\alpha = (K + \lambda I)^{-1} y, \quad \text{where} \quad K = (K)_{ij} = k(x_i, x_j) \in \mathbb{R}^{2m \times 2m},$$

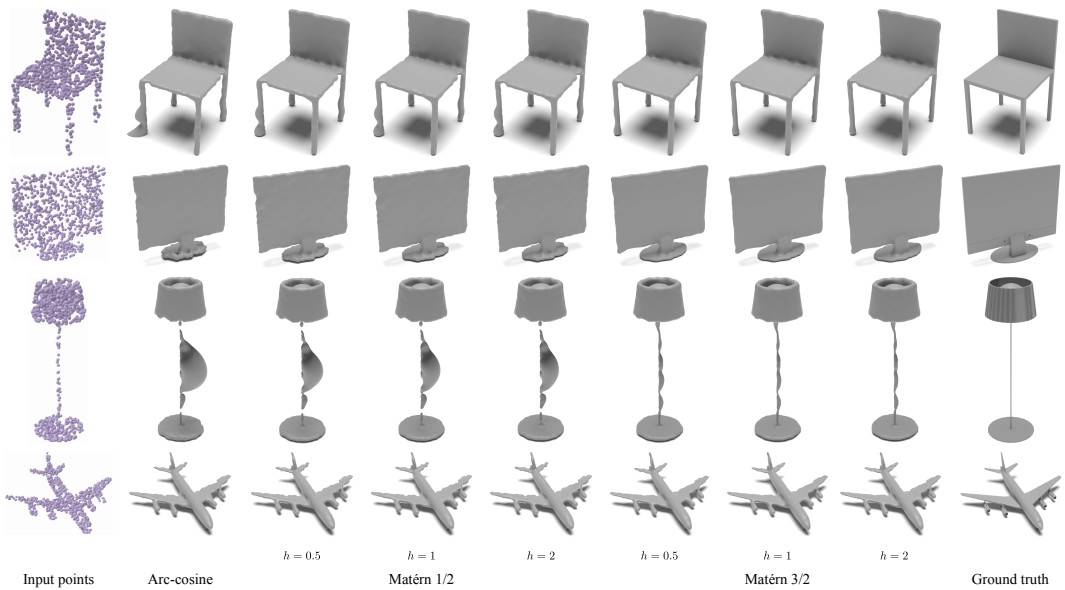

Figure 9: **Additional qualitative results on ShapeNet for non-learnable kernels**. We compare the arc-cosine kernel against Matérn 1/2 and 3/2 for different values of $h$.

and

$$y_i = \begin{cases} +\epsilon & \text{if } x'_i \in \mathcal{X}^+, \\ -\epsilon & \text{if } x'_i \in \mathcal{X}^-, \end{cases} \quad \text{for } i = 1, 2, \dots, 2m.$$

Here, $I \in \mathbb{R}^{2m \times 2m}$ denotes the identity matrix. We employ PyTorch's built-in Cholesky solver to numerically stable solve for $\alpha$ and set $\epsilon = 0.005$ for all experiments.

### F.2 FURTHER QUALITATIVE RESULTS

We show additional qualitative results on ShapeNet in Figure 9.

## G FURTHER DETAILS ON THE SURFACE RECONSTRUCTION BENCHMARK

This section details the experimental setting that has been used to run *Neural Splines* (NS; Williams et al. (2021)) on SRB in Section 4.1 of the main paper. Moreover, we present per-object metrics and additional qualitative results.

### G.1 EXPERIMENTAL SETTING

For NS, we used the official implementation provided here: `https://github.com/fwilliams/neural-splines`. Similar to the original paper, we used 15,000 Nyström samples and did the same parameter sweep over the regularization parameter, $\lambda \in \{0, 10^{-13}, 10^{-12}, 10^{-11}, 10^{-10}\}$. For the rest of the parameters (not mentioned in the paper), default values provided in the repository have been used, except for the grid size which we set to 512. We used exactly the same setting for Matérn kernels, except that we did a modest parameter sweep over the bandwidth parameter, $h \in \{0.5, 1, 2\}$. We utilized the SRB data from here: `https://github.com/fwilliams/deep-geometric-prior`.

Despite our best efforts, we were unable to reproduce the results for the NS kernel reported in the original paper. We run on four different GPUs, including NVIDIA RTX A2000, A40, A100, and V100 (which has been used by the authors of Neural Splines), and tested different configurations of the following parameters (see code): `grid_size`, `eps`, `cg-max-iters`, `cg-stop-thresh`, `outer-layer-variance`. In Table 1 of the main paper, we report the best result we could achieve on a V100, which is 0.19 for Chamfer and 3.19 for the Hausdorff distance; Williams et al.

| | Anchor | | Daratech | | DC | | Gargoyle | | Lord Quas | |
|---|---|---|---|---|---|---|---|---|---|---|
| | CD↓ | HD↓ | CD↓ | HD↓ | CD↓ | HD↓ | CD↓ | HD↓ | CD↓ | HD↓ |
| Matérn 1/2 | 0.31 | 6.97 | 0.25 | 6.11 | 0.18 | 2.62 | 0.19 | 5.14 | **0.12** | 1.32 |
| Matérn 3/2 | 0.25 | 5.08 | 0.23 | 4.90 | 0.15 | **1.24** | **0.16** | **2.54** | **0.12** | 0.90 |
| Matérn 5/2 | 0.93 | 28.33 | 0.87 | 32.17 | 0.34 | 15.98 | 0.55 | 22.52 | 0.21 | 9.49 |
| Matérn ∞ | 2.88 | 29.48 | 5.80 | 45.94 | 0.35 | 35.70 | 3.84 | 30.70 | 3.13 | 24.55 |
| NS* | 0.27 | 5.38 | 0.23 | 4.67 | 0.15 | 1.41 | 0.17 | 3.49 | **0.12** | 0.99 |
| NS | 0.22 | 4.65 | **0.21** | 4.35 | **0.14** | 1.35 | **0.16** | 3.20 | **0.12** | **0.69** |
| SPSR | 0.33 | 7.62 | 0.26 | 6.62 | 0.17 | 2.79 | 0.18 | 4.60 | **0.12** | 1.83 |
| FFN | 0.31 | 4.49 | 0.34 | 5.97 | 0.20 | 2.87 | 0.22 | 5.04 | 0.35 | 3.90 |
| Siren | 0.32 | 8.19 | **0.21** | 4.30 | 0.15 | 2.18 | 0.17 | 4.64 | 0.17 | 0.82 |
| SAP | 0.34 | 8.83 | 0.22 | 3.09 | 0.17 | 3.30 | 0.18 | 5.54 | 0.13 | 3.49 |
| DiGS | 0.28 | 5.71 | **0.21** | 5.02 | 0.15 | 2.13 | **0.16** | 3.81 | **0.12** | 1.10 |
| VisCo | **0.21** | **3.00** | 0.26 | 4.06 | 0.15 | 2.22 | 0.17 | 4.40 | **0.12** | 1.06 |
| OG-INR | 0.29 | 7.56 | 0.23 | **2.89** | 0.17 | 2.68 | 0.19 | 5.01 | 0.13 | 2.14 |

Table 6: **Per-object quantitative results on SRB**. "NS*" denotes the best result we could achieve by running the official implementation of NS, see Section G.1. **Bold** marks best result, underline second best.

(2021) reported 0.17 and 2.85 for Chamfer and Hausdorff distance, respectively. In order to achieve this, we had to lower `outer-layer-variance` form $10^{-3}$ per default to $10^{-5}$.

To measure runtime, we again used the same setting as in the original NS paper: 15,000 Nyström samples, and $\lambda = 10^{-11}$. Due to the absence of further information about the experimental setting in the original paper, we used default values provided in the repository for the rest of the parameters. Runtime was measured on a single NVIDIA V100, similar to Williams et al. (2021).

## G.2  Additional Results

Per-object metrics are shown in Table 6, and more qualitative results in Figure 10.

# H  Additional Details For Texture Reconstruction Experiment

This section provides implementation details and further information about the evaluation protocol used for the texture reconstruction experiment presented in Section 4.2 of the main paper.

## H.1  Implementation Details

Recall from Section 4.2 of the main paper that we are given a dataset $\mathcal{D}' = \mathcal{D} \times \{c_1, c_2, \ldots, c_m\} \subset \Omega \times \mathbb{R}^d \times \mathbb{R}^3$ with per-point RGB colors $\{c_1, c_2, \ldots, c_m\}$. In order to also reconstruct texture, we are seeking a function $f' : \mathbb{R}^d \to \mathbb{R}^4$, predicting signed distances as well as RGB values. Using notation introduced in Section F.1, we compute $\alpha' \in \mathbb{R}^{2m \times 4}$ as

$$\alpha' = (K + \lambda I)^{-1} y', \quad \text{where} \quad y' := [y, \bar{c}] \in \mathbb{R}^{2m \times 4}$$

and $\bar{c} := [c, c]^\top$ with $(c_1, c_2, \ldots, c_m) \in \mathbb{R}^{m \times 3}$. Then, we predict the function $f'$ as

$$f'(x) = \sum_{i=1}^{2m} \alpha'_i k(x, x'_i),$$

where $\alpha'_i \in \mathbb{R}^4$ denotes the $i$-th row of $\alpha'$. Clearly, $f'(x) = (f'_1(x), f'_2(x), f'_3(x), f'_4(x)) \in \mathbb{R}^4$ with $f'_1$ denoting the singed distance at $x$, and $f'_2, f'_3, f'_4$ are the component functions that represent RGB colors at a position $x$.

To extract a textured surface mesh from the predicted four-dimensional volume, we use a two-stage process. First, we apply Marching Cubes (Lorensen & Cline, 1987) to $f'_1$, and then trilinearly interpolate RGB colors at the previously predicted surface points.

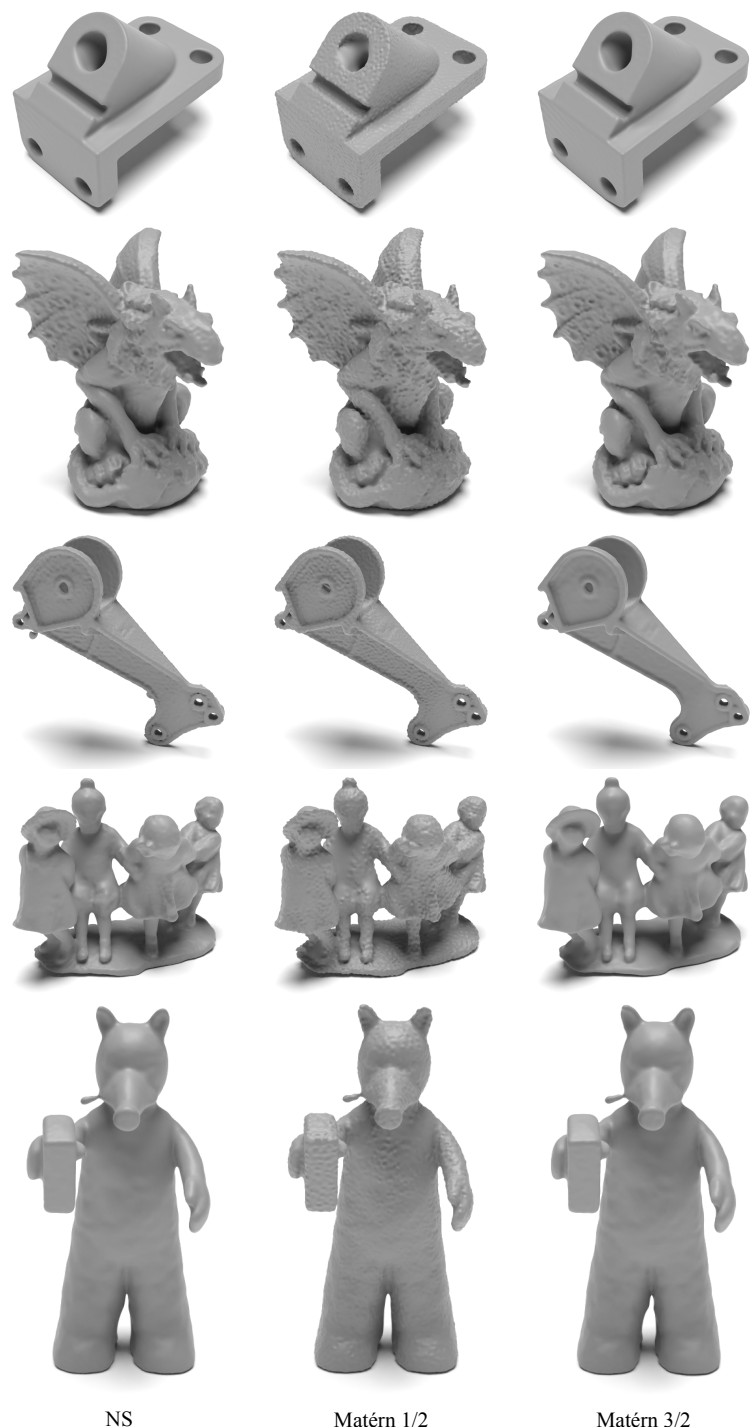

NS        Matérn 1/2        Matérn 3/2

Figure 10: **Additional qualitative results on SRB for non-learnable kernels**. We compare *Neural Splines* (NS; Williams et al. (2021)) to Matérn 1/2 and 3/2.

## H.2 Evaluation Protocol

Our quantitative evaluation on the *Google Scanned Objects* (GSO; Downs et al. (2022)) dataset closely follows Mitchel et al. (2024), in which authors compute image-based evaluation metrics on the texture atlas to quantify reconstruction ability. Specifically, we first sample $m$ surface points along with corresponding normals and per-point RGB colors from shapes in GSO. Next, we solve the KRR problem in Eq. (1) as always. Then, instead of predicting RGB colors on the estimated geometry (which generally differs between various kernel functions), we predict (*i.e.*, trilinearly interpolate) colors on the surface points of the *original* mesh. This allows us to leverage the existing UV parametrization of the original mesh to generate a texture atlas with colors predicted from KRR. Once the texture atlas at hand, we compare it to the original texture atlas using PSNR and LPIPS.

We used ten objects from GSO with different complexity. The following assets have been used:

- `ACE_Coffee_Mug_Kristen_16_oz_cup`
- `Animal_Planet_Foam_2Headed_Dragon`
- `Jansport_School_Backpack_Blue_Streak`
- `Now_Designs_Bowl_Akita_Black`
- `Predito_LZ_TRX_FG_W`
- `Reebok_ZIGTECH_SHARK_MAYHEM360`
- `Razer_Kraken_Pro_headset_Full_size_Black`
- `RJ_Rabbit_Easter_Basket_Blue`
- `Schleich_Lion_Action_Figure`
- `Weisshai_Great_White_Shark`

They can be downloaded from: `https://app.gazebosim.org/dashboard`.

## I Additional Results for Data-dependent Kernels

We show additional qualitative results for learned kernels on ShapeNet in Figure 11.

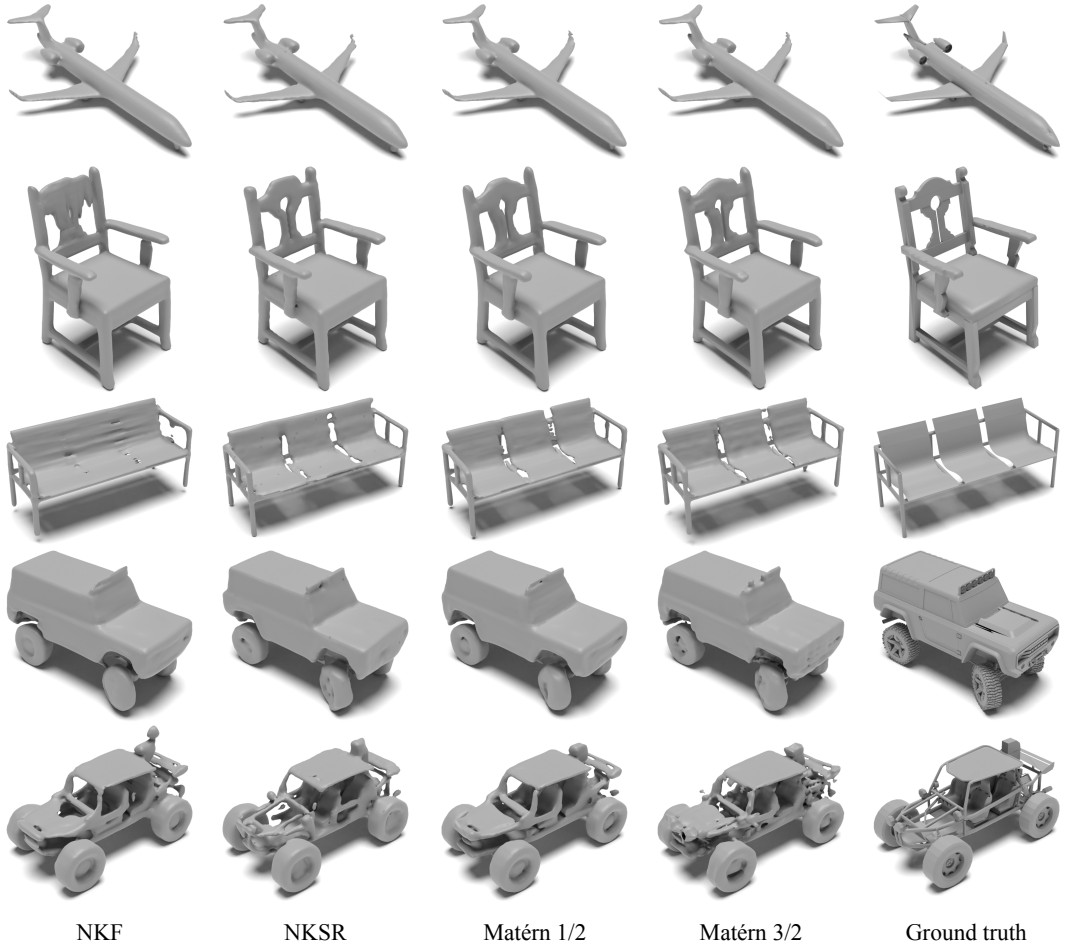

|     NKF     |    NKSR    |  Matérn 1/2  |  Matérn 3/2  |  Ground truth  |

Figure 11: **Additional qualitative results on ShapeNet for learned kernels**. We compare *Neural Kernel Fields* (NKFs; Williams et al. (2022)) and *Neural Kernel Surface Reconstruction* (NKSR; Huang et al. (2023)) against learnable Matérn 1/2 and 3/2.

