# OpenReview forum: "Matérn Kernels for Tunable Implicit Surface Reconstruction"
_ICLR.cc/2025/Conference — ICLR 2025 Poster_

### Official Review · Reviewer_QSTk · 2024-10-26

**Soundness:** 3
**Presentation:** 2
**Contribution:** 2
**Rating:** 6
**Confidence:** 4

**Summary:**

This work proposes to use Matern kernels -- a unified family of kernel functions -- for 3D reconstruction of shapes from sparsly sampled points. A key property of these kernels is tunability, in that the sharpness of these kernels can be controlled. The authors claim this can help to overcome the known implicit bias in MLPs against the reconstruction of high-frequency features. The contributions of this paper are mostly techincal. The authors provide a theoretical investigation of Matern Kernels, including several toy experiments in support of derivations.

**Strengths:**

Overall, the authors investigation into the suitability of Matern kernels for surface reconstruction in section 4 is the standout.

While the experimental details are unclear, the analysis of the eigenvalue decay exposes the mechanics of how the kernels can address spectral bias and supports the authors claims.

The discussion of the reconstruction error is compelling, and the highlight of the paper. The authors cleverly derive an upper bound on the reconstruction error in terms of the kernel parameters (h, $\nu$), and expose a canonical threshold (Equation 13).

To the best of my knowledge, the mathematical formulations are sound, though I did not check them closely.

**Weaknesses:**

1). Despite the contribution of the paper being technical, the way the math is introduced and discussed is unintuitive and confusing. Section 3 contains significant mathematical "filler" e.g. equations (5-8) could be moved to the appendix.  The RKHS norm is introduced early without context. Only a page later, in Equation (12), does its purpose become clear in supplying the upper bound for the reconstruction loss. Even then, it is not clear that it is necessary to reproduce the explicit form of the RHKS norm in the main text (this can be moved to the supplement).  I question the necessity of section 3, as none of the math introduced is the authors' own work and makes it less clear what their contributions are. I think most of it can be removed, and combined with section 4, with the appropriate math introduced in the main text only as it relates to Eqs. 12-13 and the rest left to the supplement.

Additionally, I'm on the fence about whether 4.2 should be kept in the main paper or moved to the supplement. These insights are nice to know, but don't necessarily have any effect on the implementation, unlike the discussion in 4.1.

The extra space gained by condensing the above should give the authors more room to both A). make clear exactly what their own contributions are (e.g. the derivations of the upper bound, etc.) and B). provide more room for experiments and discussion.

2). The biggest weakness of the paper are the experimental results. Overall, the authors proposed approach performs only comparably relative to competing methods and does not distinguish itself. This is unfortunate, because the insights in section 4.1 are very nice and had me rooting for the method, but they don't seem to provide a significant increase in performance over existing methods. Overall, the proposed method is not substantially different from prior methods like NKF & NKSR in that it is using just a different choice kernel for surface reconstruction. In general, this is OK, and the authors provide a very nice theoretical investigation into the kernel properties, but in such a case its necessary to show unambiguous, generally uniform improvements over previous similar methods to justify the approach.

Another important question left unanswered is scalability. Experiments are shown in which simple surfaces are reconstructed from 1000 points. How does the kernel reconstruction perform from sparse samples of 10K or 100K points? I am concerned that the Cholesky factorization is not scalable in these cases, though this does not seem fundamental to the method and the CG could be used. It should be noted that JAX has a very scalable, efficient, and differentiable implementation of the CG solve for sparse systems, and this may be of interest to the authors.

In a similar vein, no results are shown comparing performance with sampling irregularity. How does the method perform with uniformly distributed samples vs varying degrees of non-uniformly sampled points?

Together, scalability and robustness under sampling irregularity are two potential axes for comparison against other methods that could allow the proposed method to more obviously stand out. Additionally, what about reconstructing a complex texture on a surface (e.g. from an Objaverse mesh)? A complex texture is itself a high-frequency signal that could help to distinguish the authors approach.

**Questions:**

In the eigenvalue decay experiments, the authors should state explicity they are considering the eigenvalues of the kernel K(x, y). Assuming I understand this correctly, how many points are used and how are they distributed in this experiment? What shapes (if any) do they represent?

Can h* in equation 13 be computed or estimated from the sampled points so as to give an exact way to recover the optimal h value given $\nu$?

---

> ### Author Response · Authors · 2024-11-21
>
> Dear Reviewer QSTk,
>
> thank you for the professional evaluation of our manuscript and your valuable comments. We tried our best to address your concerns and hope we could clarify them below and in the updated version of the paper. Please reach out to us if you still have open questions or points that need further clarification. We’re happy to answer them.
>
> **"Despite the contribution of the paper being technical, the way the math is introduced and discussed is unintuitive and confusing“:**
>
> We agree with most of the points and are grateful for the feedback. We revised the paper accordingly and think that this greatly improved readability. Specifically, as suggested, we removed Section 3 and merged relevant parts of it with Section 4. The RKHS norm is now introduced after Eq. (12) (now Eq. (10)) on page 5. We decided to leave the explicit form in the main paper as we later study the norm’s dependence on $h$—something that is not entirely obvious without having the explicit form at hand. We also decided to leave Eqs. (5)-(8) in the main paper as it might be important to understand some of the benefits of Matérn kernels (e.g., easy implementation), and other reviewers actually appreciated the self-containedness of the paper.
>
> **"Additionally, I'm on the fence about whether 4.2 should be kept in the main paper or moved to the supplement“:**
>
> We agree. However, we see this section more as an additional justification for why we believe Matérn kernels are a good choice for kernel-based surface reconstruction. Specifically, Section 4.2 (now Section 3.3) is meant to put Matérn kernels more into context, relating them to neural networks and the arc-cosine kernel, and convincing the reader that we didn’t just pick them „randomly“.
>
> **"The extra space gained by condensing the above should give the authors more room to both A) make clear exactly what their own contributions are and B) provide more room for experiments and discussion“:**
>
> As suggested, we added an explicit list of contributions to the introduction, see L115-121. We also added more experiments (see Section 4.2) and comparisons to more recent related work (see Table 1 and also comments for Reviewer 57Rh).
>
> **"The biggest weakness of the paper are the experimental results. Overall, the authors proposed approach performs only comparably relative to competing methods and does not distinguish itself. […] It is necessary to show unambiguous, generally uniform improvements over previous similar methods to justify the approach“:**
>
> We agree that the proposed method is not substantially different from NS and NKF in that we (from a practical perspective) "just“ exchanged the kernel function. This is something we are aware of, and there is not much to argue about. However, we do want to add the following comments:
> - While from a practical perspective, we more or less just exchanged a kernel function, we do contribute important results on the theory side. Most notably, Theorem 3. ReLU networks, already related to kernel methods, and SIRENs are two widely used architectures—by relating SIRENs to kernel methods, we close an open gap (which has already been posed in [1]). Moreover, the presented proof is elegant, builds upon Bochner’s theorem and leverages modern NTK theory. Besides Theorem 3, we state a new bound of the reconstruction error (Proposition 2), which gives practical insights on how to effectively tune Matérn kernels. Theorem 3 and Corollary 5 may also be of interest to researchers working on machine learning theory, trying to understand the connection between kernel methods and neural nets.
> - While Matérn kernels do not lead to a massive boost in reconstruction accuracy, they do *consistently* improve previous methods as shown in Tables 1 and 2. The only exception is NKSR. One of the big advantages of Matérn kernels is that they are significantly faster to compute and train. As such, while not surpassing the reconstruction accuracy of NKSR (difference in IoU of only 0.3, though), learnable Matérn kernels lead to a five times shorter training time than NKSR. The goal of this paper was not to present a method that surpasses NKSR; NKSR is well-engineered, highly optimized, and challenging to implement. From a practical perspective, the aim of this paper was to show that a simple change of kernel function can lead to improved performance and massive speedup. Both is demonstrated in Section 4 of our paper.

---

> ### Author Response · Authors · 2024-11-21
>
> **"Another important question left unanswered is scalability. […] How does the kernel reconstruction perform from sparse samples of 10K or 100K points?“:**
>
> Scalability is always a concern for kernel methods in general, and we agree (and are aware) that the employed Cholesky factorization used within the ShapeNet experiment does not scale to a large number of input points. To make Matérn-based 3D reconstruction scalable, we closely followed the NS framework. In brief, we use Nyström sampling, an efficient GPU-based CG solver (FALKON), and, optionally, allow for a chunked reconstruction by subdividing the input domain into individual cells and then reconstructing each cell separately. This implementation has been used for the SRB experiment (as mentioned in Section 4.1, starting L420); it is able to reconstruct an almost 100K point cloud in under 10 seconds. As shown in the NS paper for the arc-cosine kernel, the same implementation in fact scales to even larger input point clouds with more than nine million points.
>
> We were not aware of the sparse Cholesky implementation in JAX—this is an extremely helpful pointer. Thank you!
>
> **"How does the method perform with uniformly distributed samples vs varying degrees of non-uniformly sampled points?“:**
>
> We are currently not sure how to implement an experiment in which we sample points with *varying degrees of non-uniformity*. Generally, for all experiments, points are sampled uniformly. The only exception is SRB because these point clouds do contain holes and missing parts, effectively modeling „non-uniformity“. We are still thinking about how to design this experiment and will post an update within the next three days as we would really love to include this experiment. In case you have any suggestions, we would love to hear them. Thank you for your patience.
>
> **"Additionally, what about reconstructing a complex texture on a surface (e.g. from an Objaverse mesh)?“:**
>
> This is an excellent idea! We added an experiment in which also reconstructed texture, please see Section 4.2. Matérn kernels indeed lead to improved texture reconstructions with fewer artifacts, sharper details, and visually more pleasing textures. Note: As we are not yet happy with the layout of the added Figure 7, it might change within the next three days. Also, detailed implementation details in Appendix F are still missing; we will update them within the next three days as well. We will leave a comment here once done. Thank you for your patience.
>
> **"In the eigenvalue decay experiments, the authors should state explicity they are considering the eigenvalues of the kernel $k(x, y)$. Assuming I understand this correctly, how many points are used and how are they distributed in this experiment? What shapes (if any) do they represent?“:**
>
> The EDR plotted in Figure 3 is the *theoretical* decay rate obtained from Theorem 1.
>
> **"Can $h^\*$ in Eq. (13) be computed or estimated from the sampled points so as to give an exact way to recover the optimal $h$ value given?“:**
>
> This is a very interesting question. Estimating $h^*$ essentially boils down to estimating $Q$ in Eq. (12). Assuming a fixed Matérn kernel (fixed $\nu$) and point dimension, $d$, $Q$ only depends on the Fourier transform of the unknown SDF, $f$. Assuming the input points, say $X$, are sampled from the true surface, we do know that $f[X]=0$. The problem is that the remaining values of $f$ are still unknown, which makes it impossible to compute $h^*$ from just sampled points. However, we could envision that $h^*$ may at least be approximated if, along with on-surface samples, some off-surface points with corresponding signed distances would be available. We leave this for future work as it is itself a very hard question, but agree that this is a very interesting route to proceed.
>
> If you feel we still did not properly address any of your concerns, please let us know. Thank you again for your efforts to evaluate our manuscript.
>
> ---
>
> [1] Williams et al. Neural Splines: Fitting 3D Surfaces with Inifinitely-Wide Neural Networks. CVPR, 2021.

---

> ### Comment · Reviewer_QSTk · 2024-11-21
>
> Thanks for your thorough and comprehensive rebuttal. More generally I appreciate the considerable effort it must have taken the authors to add the somewhat extensive additional experimental results reported in the rebuttal.
>
> That said, I'm quite pleased with the results and the author response. I appreciate the restructuring of the first few sections to better highlight the authors contributions. While I disagree with the other reviewers and believe the authors could go further in removing spurious equations, I think the revisions are a suitable compromise.
>
> I also think the paper is strengthened by the additional comparisons to SoTA surface reconstruction methods detailed in the response to reviewer 57Rh.
>
> The additional experiments on texture reconstruction are also very much appreciated. If, in the interim before publication, the authors could additionally present quantitative results (e.g. using the metrics + similar collection of textured shapes as in the experiments in Table 1 of https://arxiv.org/pdf/2312.09250 - though no need to compare as an entirely different method) this would be a difference maker for me and tip to recommend accept.  Additionally, a key component of this experiment I think would be illuminating would be to show the rate at which the metrics decay as the number of sampled points decreases. 200K points is quite high, and I'm very curious how the proposed kernels affect the quality of the reconstructed with limited data.
>
> I also thank the authors for their willingness to consider discretization/sampling experiments. As an example, the authors might consider evaluating the method against competing approaches after resampling in a similar manner as is done in https://github.com/nmwsharp/discretization-robust-correspondence-benchmark. Alternatively, lossy real-world data could be tried (e.g. point clouds of a scene).  That said, considering the effort the authors have already spent on the rebuttal and additional experiments, I would consider this a "nice to have" but not a necessary condition for acceptance.
>
> As a general comment, I agree with the authors that their technical contributions are, in fact, the primary contributions of the paper. I think that the updated experimental results are now sufficient to support the paper on its technical merits. That said, I think the authors should better highlight the efficiency and ease of the Matern implementation (perhaps with some pseudocode showing the "two-line" implementation") relative to other methods, potentially working this into the clearly stated contributions in the intro.  I think the statement in the rebuttal
>
> "The goal of this paper was not to present a method that surpasses NKSR; NKSR is well-engineered, highly optimized, and challenging to implement. From a practical perspective, the aim of this paper was to show that a simple change of kernel function can lead to improved performance and massive speedup."
>
> is a great encapsulation of the contributions and should be worked into the intro in some form (though not necessarily having to mention the NKSR0.
>
> As a last point to clarify my previous comment, JAX has a very efficient sparse conjugate gradient (CG) solver (https://jax.readthedocs.io/en/latest/_autosummary/jax.scipy.sparse.linalg.cg.html), not Cholesky solver.
>
> Based on the above comments I am willing to raise my score to weak acceptance.

---

> ### Author Response · Authors · 2024-11-25
>
> Dear Reviewer QSTk,
>
> thank you for the prompt reply. We very much appreciate that you have raised your score and are super happy that we could clarify most of your concerns. To address your remaining concerns and to incorporate your feedback, we made the following edits to the paper:
>
> - We removed Eq. (5). We agree that this equation is somewhat spurious, and you are right, it is in fact not needed to understand the paper (we also believe it does not affect the self-containedness of the paper).
> - We added a quantitative evaluation for the texture reconstruction experiment on Google Scanned Objects (GSO) in Section 4.2, closely following the evaluation protocol from the paper [1] you sent to us. First, we sampled $m$ surface points along with corresponding normals and per-point RGB colors from shapes in GSO. Next, we solved the KRR problem in Eq. (1) as usual. Then, instead of predicting colors on the estimated geometry, we predicted (i.e., trilinearily interpolated) colors on the surface points of the *original* mesh. This way we could leverage the existing UV parametrization of the original mesh to generate a texture atlas with colors predicted from the kernel (also, this step is crucial to ensure a fair comparison across different kernels—computing a UV parametrization from the predicted geometry would be unfair since predicted geometry generally differs between various kernel functions). As requested, we also evaluated how metrics behave if the number of sampled surface points vary (for $m=10,000$ and $m=2,500$). We evaluated Matérn 3/2 (the best-performing Matérn kernel) against the arc-cosine kernel and observe a consistent improvement in both settings, dense and sparse. Notably, we did not tune Matérn kernel’s bandwidth parameter (we set $h=0.1$ empirically). We took 10 objects with varying degrees of complexity from GSO for this evaluation (which is in a similar magnitude as the evaluation in [1]). Lastly, we would like to briefly argue why we did not include a comparison to [1] in the first place (since we are not sure whether you expected us to add such comparison). First off, compared to [1], we did not re-mesh and re-parametrize original objects to obtain high-res (30K) and low-res (5K) versions of the dataset. Instead, we simply took the original resolution of the mesh together with the supplied UV parametrization. The only reason for this is the limited time frame we had to implement and carry out experiments. However, as the resolution *across kernels* is the same, we believe our evaluation is fair, but simply not directly comparable to [1]. Second, even if we did use the same resolutions as in [1], it still would have been an unfair comparison as [1] relies on *pre-trained* VAEs (their "FL-VAE“) on large image datasets. In conclusion, while not perfect, we believe the added quantitative comparison nicely shows the benefits of Matérn kernels while being carried out thoroughly. We will update Appendix H within the next day.
> - We tried to incorporate your feedback regarding the „encapsulation of the contributions“. We updated L96-99, trying to make it clearer that the contributions of this paper are of practical *and* theoretical nature. We also added L124-126.
>
> More generally, we are still not sure how to incorporate pseudo-code as the CUDA implementation of the arc-cosine kernel is lengthy. We would like to note that we will publish the source code, and might just refer to it. The code for, e.g., Matérn 1/2 we use is:
> ```
> d_xy = torch.cdist(x, y)
> k_12 = torch.exp(-d_xy / h)
> ```
> which could be squeezed even into a single line. The official CUDA implementation of the arc-cosine kernel can be found here: https://github.com/fwilliams/neural-splines/blob/main/neural_splines/falkon_kernels.py, L98-145.
>
> Furthermore, we will try to incorporate a small experiment regarding sampling irregularities as the repository you are referring to already provides the respective data (we would just have to run our methods). We will try our very best to include this experiment in time. Thank you for pointing us to this repository. This would be an interesting experiment which we will definitely carry out in the future.
>
> Finally, we would like to mention again that we truly appreciate your honest, fair, and constructive feedback. This really helped us improve the paper. If you still have open questions, please let us know. We are happy to answer.
>
> ---
> **Edit (11/26):** We have updated Appendix H. It now includes implementation details for the texture reconstruction experiment as well as information about the evaluation protocol.
>
> ---
> [1] Mitchel et al. Single Mesh Diffusion Models with Field Latents for Texture Generation. CVPR, 2024.

---

> > ### Author Response · Authors · 2024-11-30
> >
> > Dear Reviewer QSTk,
> >
> > this is a follow-up comment to inform you that we have carried out the discretization/sampling irregularity experiment you were suggesting. We used all meshes from the dataset you mentioned (20 meshes for each sampling strategy, 80 in total), which can be found here: https://github.com/nmwsharp/discretization-robust-correspondence-benchmark. The dataset includes meshes created according to the following four sampling strategies (cited from the repository):
> >
> > - `dense` Meshes are refined to have nonuniform density by choosing 5 random faces, refining the mesh in the vicinity of the face, then isotropically remeshing.
> > - `iso` Meshes are isotropically remeshed, to have a roughly uniform distribution of vetices, with approximately equilateral triangles
> > - `mc` Meshes are volumetrically reconstructed, and a mesh is extracted via the marching cubes algorithm.
> > - `qes` Meshes are first refined to have many more vertices, then simplified back to approximately 2x the original resolution using Quadric Error Simplification
> > - `cloud` A point cloud, with normals, sampled uniformly from the mesh
> >
> > We used point clouds from `cloud` as ground truth to compute metrics, F1-Score (first table) and Chamfer distance (second table). Notable, we did not use Nyström sampling for this experiment (to fully rely on the actual data) and set $h=0.5$.
> >
> > |           | dense     | iso       | mc        | qes       |
> > |-----------|-----------|-----------|-----------|-----------|
> > | Matérn 3/2 | **79.9**    | **95.3**      | **91.0**      | **97.9**   |
> > | Arc-cosine | 79.7      | 95.1      | 90.7      | 96.9   |
> >
> >
> > |           | dense     | iso       | mc        | qes
> > |-----------|-----------|-----------|-----------|-----------|
> > | Matérn 3/2  | **4.81**  | **4.05**  | **4.15**  | **3.50**  |
> > | Arc-cosine  | 4.98  | 4.26  | 4.37  | 4.29  |
> >
> > We see that Matérn 3/2 consistently outperforms the arc-cosine kernel, where the biggest difference is measured for the `qes` sampling strategy. This suggests that Matérn kernels can deal much better with irregular data than the arc-cosine kernel. We believe this is a very interesting finding (which highlights the benefits of Matérn kernels one more) and would like to thank the reviewer again for suggesting this experiment. We unfortunately can't revise the paper at this stage, but we will include the results in the revised version of the paper.

---

### Official Review · Reviewer_57Rh · 2024-11-01

**Soundness:** 3
**Presentation:** 3
**Contribution:** 3
**Rating:** 6
**Confidence:** 3

**Summary:**

This paper proposes using the family of Matérn kernels for implicit surface reconstruction, targeting the 3D reconstruction of oriented point clouds. Leveraging the recent advances in kernel-based reconstruction, the authors demonstrate that Matérn kernels possess theoretical and practical advantages that make them highly suitable for surface reconstruction tasks. They argue that these kernels surpass state-of-the-art methods based on the arc-cosine kernel, offering improved simplicity, computational efficiency, and scalability.

Key contributions include:
1.Highlighting the Matérn kernels’ stationary properties, which allow tunable spectral properties similar to Fourier feature mappings, addressing spectral bias issues in coordinate-based MLPs.
2.Providing theoretical insights into the Matérn kernel’s relationship with SIREN networks and its distinctions from arc-cosine kernels previously used in surface reconstruction.
3.Introducing data-dependent Matérn kernels based on Neural Kernel Fields and showing that the Laplace kernel (a member of the Matérn family) performs competitively with state-of-the-art methods in noise-free settings while significantly reducing training time.

**Strengths:**

(1) The paper presents a tunable implicit surface reconstruction method based on Matérn kernels. The tunability of the method allows users to adjust parameters based on specific application needs, optimizing reconstruction performance in different scenarios, which is crucial in practical applications.
(2) Matérn kernels are widely used in statistics and machine learning, and the paper effectively leverages their favorite mathematical properties, making the proposed method more rigorous and interpretable theoretically.
(3) The paper is well-structured and logically coherent, making it easier for readers to understand the methods and results. The theoretical proofs and experiments in this paper are relatively sufficient.

**Weaknesses:**

(1) Lacks comparison with the latest implicit surface reconstruction algorithms.
(2) Some figures are missing ground truth.

**Questions:**

(1) SIREN is a work from 2020, and there have been many recent improvements based on SIREN. The authors should consider these in their comparative experiments to showcase the advantages of Matern kernels in implicit reconstruction.
(2) In the sparse surface reconstruction, in addition to quantitative results, including visualizations of the sampled points would make the effects more intuitive.

---

> ### Author Response · Authors · 2024-11-21
>
> Dear Reviewer 57Rh,
>
> thank you for the professional evaluation of our manuscript and your valuable comments. We tried our best to address your concerns and hope we could clarify them below and in the updated version of the paper. Please reach out to us if you still have open questions or points that need further clarification. We’re happy to answer them.
>
> **"Lacks comparison with the latest implicit surface reconstruction algorithms“ and "The authors should consider these in their comparative experiments to showcase the advantages of Matern kernels in implicit reconstruction“:**
>
> We added comparisons with state-of-the-art methods on the challenging SRB as this dataset contains real-world scans (including holes and incomplete data) thus allowing for the most informed conclusions about the true practical performance of Matérn kernels. Specifically, we added SAP [1], DiGS [2], VisCo [3], and OG-INR [4]. To ensure fair comparison, all chosen methods are optimization-based (i.e., no learned prior is involved) *and* use surface normals; Matérn 3/2 outperforms each of the five methods as seen in the updated Table 1. We also updated the SRB paragraph, see L417-433.
>
> We also plan to add more comparisons to state-of-the-art methods for *learnable* kernels. We are currently awaiting results for representative ("post-SIREN“) methods such as POCO [5], ALTO [6], and DITTO [7]. We will update the paper within the next three days and, once updated, will leave a comment here. Thank you for your patience!
>
> **"Some figures are missing ground truth“ and „In the sparse surface reconstruction, in addition to quantitative results, including visualizations of the sampled points would make the effects more intuitive“:**
>
> We will add the sparsely sampled input points to Figure 2 and Figure 6 (right) as overlays to reconstructed surfaces. Besides, we will add the ground truth surfaces to Figure 6 (right). We will also update figures in the Appendix. We will do so within the next three days, and, once done, will leave a comment here. Thank you again for your patience.
>
> If you feel we still did not properly address any of your concerns, please let us know. Thank you again for your efforts to evaluate our manuscript.
>
> ---
>
> [1] Peng et al. Shape As Points: A Differentiable Poisson Solver. NeurIPS, 2021.
>
> [2] Ben-Shabat et al. DiGS : Divergence guided shape implicit neural representation for unoriented point clouds. CVPR, 2022.
>
> [3] Pumarola et al. VisCo Grids: Surface Reconstruction with Viscosity and Coarea Grids. NeurIPS, 2022.
>
> [4] Koneputugodage et al. Octree Guided Unoriented Surface Reconstruction. CVPR, 2023.
>
> [5] Boulch et al. POCO: Point Convolution for Surface Reconstruction. CVPR, 2022.
>
> [6] Wang et al. ALTO: Alternating Latent Topologies for Implicit 3D Reconstruction. CVPR, 2023.
>
> [7] Shim et al. DITTO: Dual and Integrated Latent Topologies for Implicit 3D Reconstruction. CVPR, 2024.

---

> ### Author Response · Authors · 2024-11-25
>
> Dear Reviewer 57Rh,
>
> this is a follow-up comment to inform you about the changes we promised earlier and have now implemented.
>
> - We have updated Figures 2 and 6. Figure 2 now shows the sampled input points. We also added ground truth to Figure 6.
> - We added missing per-object metrics for newly evaluated methods (SAP, DiGS, VisCo, OG-INR) on SRB to Table 6 in the Appendix.
> - We completed the comparison of learnable Matérn kernels to kernel-free methods POCO, ALTO, and DITTO. To do so, we took the best performing Matérn kernel (Matérn 1/2) and trained it with noisy (standard deviation of 0.005) input points to make our results comparable to POCO, ALTO, and DITTO. However, we have to admit that we completely oversaw that neither of the kernel-free methods (POCO, ALTO, DITTO) uses ground truth normal information. Since our method does use normals, our comparison would not be fair. We then tried to train POCO with normal information (as there is code available), but could not get a well-performing model (we found that this seems to be a common issue; authors of NKSR reported the same problem in their paper, see page 16). Ultimately, we decided not to include this evaluation in the paper, as it simply would be unfair as we think. We truly apologize for the false promise and hope that the added comparisons for SAP, DiGS, VisCo, and OG-INR on the real-world SRB did strengthen your trust in our evaluation. If this is not the case, please let us know. We are happy to include your suggestions. For completeness, we do report here the results of our evaluation:
>
> |           | IoU ↑  | CD ↓  | NC ↑  |
> |-----------|--------|-------|-------|
> |  Matérn 1/2  | 91.4 | 3.0 | 94.6|
> | POCO      | 88.4   | 4.0   | 92.8  |
> | ALTO      | 90.5   | 3.5   | 94.0  |
> | DITTO     | 92.6 | 3.2 | 94.9 |
>
> We will update the remaining Figures 9 and 11, as promised, in the Appendix within the next two days. Thank you again for your consideration and the evaluation of our manuscript.
>
> ---
> **Edit (11/27):** We have updated Figures 9 and 11 with input points and ground truth. We decided to not show input points in Figures 6 and 11 (qualitative results for learnable kernels) due to space constraints in Figure 6 (adding an extra column would have made the figure too small—for consistency reasons, we also decided to not show input points in Figure 11). We believe adding input points and ground truth greatly improved all figures and would like to thank you again for this reminder.

---

### Official Review · Reviewer_JfMa · 2024-11-04

**Soundness:** 3
**Presentation:** 4
**Contribution:** 2
**Rating:** 8
**Confidence:** 3

**Summary:**

The paper promotes the use of the Matérn kernel family for implicit surface reconstruction, emphasizing the kernel’s adaptability through two parameters that control its tendency to fit high-frequency details. These claims are substantiated through both theoretical analysis and empirical evidence. The approach is validated on standard surface reconstruction benchmarks.

**Strengths:**

The paper is well-organized and clearly written, with a sufficient amount of background material to make it self-contained.

The incorporation of Matérn kernels appears to be a valuable contribution, supported by theoretical analysis on their ability to capture high-frequency details effectively.

Both quantitative and qualitative comparisons are provided, highlighting key properties of the proposed method.

**Weaknesses:**

Unclear Benefits of Matérn Kernel Properties

Section 3.1 describes Matérn kernel properties, yet it’s unclear how differentiability and stationarity specifically benefit surface reconstruction. More specifically,
  Differentiability: Should the differentiability of the kernel be related to the (unknown) ground truth function of the surface reconstruction? what are some typical cases/assumptions, for which the kernel differentiability class is appropriate?
Stationarity: How does stationarity improve reconstruction in this context? Some empirical evidence would help clarify its usefulness. Additionally, given the local support property, wouldn’t we expect symmetric parts, such as the chair legs in Figure 6, to yield consistent reconstruction results?

The Role of Local Support

Since the product of two kernels remains a kernel, could any kernel be multiplied by a locally supported kernel to achieve local support? Why is this a specific property of the Matern kernel? NKSR, for example, multiplies the dot product kernel by a locally supported kernel. Additionally, would using a kernel with a tunable support radius (like NKSR) allow for similar control over high-frequency detail? If so, the added value of using a two-parameter Matérn kernel remains unclear.

Comparison Quality

Comparison with Arc-Sine Kernels. The chosen baseline is the arc-sine kernel. Why not compare it to the dot-product kernel with local support, as used in NKSR?

Comparison with Gaussian Kernel. In Table 1, the Gaussian kernel $(\nu = \inf)$ is presented as inferior. However, how does a two-parameter kernel like  $\nu \exp(-\tau^2/h)$  perform? What specific benefits does a Matérn kernel offer over this one? Is there an advantage in tuning Matérn kernel parameters over this form?

Unclear Connection to SIREN

Theorem 3 seems more like a property of SIREN than the Matérn kernel. It appears that the spectral density of any (stationary) kernel could satisfy the conditions in line 329, so the theorem might hold more generally than just for Matérn kernels. Could the relevance of this theorem to the Matérn kernel specifically be clarified?

**Questions:**

I would appreciate any response to the questions stated above.

---

> ### Author Response · Authors · 2024-11-21
>
> Dear Reviewer JfMa,
>
> thank you for the professional evaluation of our manuscript and your valuable comments. We tried our best to address your concerns and hope we could clarify them below and in the updated version of the paper. Please reach out to us if you still have open questions or points that need further clarification. We’re happy to answer them.
>
> **"Should the differentiability of the kernel be related to the (unknown) ground truth function of the surface reconstruction?“ and "What are some typical cases/assumptions, for which the kernel differentiability class is appropriate?“:**
>
> The reconstructed surface and the unknown ground truth surface both inherit the differentiability class of the kernel function (since we assume that the true surface is part of an RKHS). This is a consequence of the reproducing property which applies to an RKHS. A benefit of having explicit control over the smoothness of the kernel is that in the presence of noise, one can intentionally choose a smoother kernel (a higher value for $\nu$) which leads to a smoother, more regularized surface reconstruction that is less sensitive to noise (demonstrated in Section 4.3 (in the updated version) for data-dependent Matérn kernels). Furthermore, if the roughness of the shapes to be reconstructed is known in advance, one can choose the smoothness of the kernel accordingly, thus allowing to include domain knowledge (as inductive bias) into the surface reconstruction problem. This is not possible with the arc-cosine kernel. Furthermore, the only way to make the arc-cosine kernel more robust against noise is by applying stronger regularization to the KRR problem, see Eq. (1). Varying $\lambda$ in Eq. (1) is far less intuitive for users than varying a parameter, $\nu$, that controls the smoothness of a kernel (a property that can also be visualized, see Figure 1). We updated the respective paragraph (L188-195) in the paper.
>
> **"How does stationarity improve reconstruction in this context? Some empirical evidence would help clarify its usefulness“:**
>
>  Since Matérn kernels are stationary (i.e., translation invariant), their value only depends on the difference between two points. The absolute positions (the embedding) are irrelevant. Hence, the reconstructed surface remains always exactly the same, no matter where the input point cloud is located in space (translating the point cloud does not change the reconstructed surface). This is not the case for the arc-cosine kernel, as it is not translation invariant. Changing the absolute position of the point cloud also changes the reconstructed surface. Note that in order to obtain translation invariance for the arc-cosine kernel, one always has to apply a pre-processing step in which an affine transformation is estimated to ensure a zero mean of the input point cloud. We updated the respective paragraph (L196-203) in the paper and added a visualization to Appendix A (Appendix A, including Figure 8, is still very rough and will be updated within the next three days).
>
> **"Additionally, given the local support property, wouldn’t we expect symmetric parts, such as the chair legs in Figure 6, to yield consistent reconstruction results?“:**
>
> The reconstructed surface of the chair legs in Figure 6 (now Figure 9) is not symmetrical because the input point cloud is not symmetrical.
>
> **"Additionally, would using a kernel with a tunable support radius (like NKSR) allow for similar control over high-frequency detail?“:**
>
> This is a good yet very tough question. NKSR uses a (locally supported) dot-product kernel, which is not stationary. Hence, Bochner’s theorem does not apply—a tool that relates kernel functions to their spectral characteristics through a very simple mathematical transformation: the Fourier transform. As a consequence, none of the arguments provided in previous works [1,2] can be used to study the explicit control over frequencies nor spectral bias.
>
> There is theory [3] that allows computing spectral densities of non-stationary kernels, albeit more complex. We have to admit that, currently, it is unclear to us how a tunable support radius of a dot-product kernel depends on its spectrum. The steps that need to be taken would be to first write down the kernel function (currently, NKSR’s kernel formulation does not involve an explicit parameter to control the "support radius“). Then, one needs to analytically compute the spectrum of the kernel, and finally study its dependence on the support radius. At first glance, this seems to be quite challenging, without having much of a benefit: not only would the method be much more complex, harder to understand, and less intuitive, nor would it fit any of the previously presented arguments [1,2]. Nevertheless, we will continue to think about this.

---

> ### Author Response · Authors · 2024-11-21
>
> **"Since the product of two kernels remains a kernel, could any kernel be multiplied by a locally supported kernel to achieve local support? Why is this a specific property of the Matern kernel?“:**
>
> Theoretically, any kernel can be multiplied by a locally supported kernel, and the result will be locally supported. From a practical perspective, however, this makes only sense for kernel functions with spatial decay (such as Matérn kernels), i.e., kernels for which $k(x,y)\rightarrow 0$ as $\Vert x-y\Vert\rightarrow 0$. Although the kernel itself is actually not compact (because it never becomes zero exactly), it can be considered locally supported as the tail is getting infinitely small and, hence may be trimmed away without losing information. On the other hand, if one multiplies a kernel without spatial decay (such as the dot-product or arc-cosine kernel) with a compact kernel, a lot of information will be lost, resulting in bad predictions (spurious geometry in our case).
>
> The reason why NKSR can modulate a non-compactly supported dot-product kernel and it still works is that they predict multiple hierarchies of kernels, effectively taking into account long-range dependencies.
>
> **"The chosen baseline is the arc-sine kernel. Why not compare it to the dot-product kernel with local support, as used in NKSR?“:**
>
> A (not data-depended, i.e., not learned) dot product kernel does not lead to meaningful surface reconstructions at all (and so does a locally supported dot product kernel). A dot-product kernel, being a linear kernel function, can only reconstruct planar surfaces, i.e., planes. In NKSR, however, the dot product kernel is *learned*, i.e., the input to the kernel are neural-network-transformed input coordinates, effectively lifting points to a higher dimensional space in which a dot-product kernel could describe its shape. We did compare learnable Matérn kernels to NKSR (with a locally supported dot-product kernel).
>
> **"However, how does a two-parameter kernel like $v*exp(-\tau^2/h)$ perform? What specific benefits does a Matérn kernel offer over this one?“:**
>
> This is an excellent question. Multiplying $\nu$ to a kernel function does not change its mathematical properties (differentiability, for instance). It simply increases variance by scaling the magnitude of the kernel. Specifically, when using a kernel function of the form $\nu k_h(\tau)$, it can be shown that $\nu$ scales the regularization parameter $\lambda$ in Eq. (1) by $1/\nu$. Hence, the effect of $\nu$ is marginal as instead of introducing a second parameter, one could simply adjust $\lambda$ directly. From a practical perspective, however, in both variants, i.e., for a kernel of the form $\nu k_h(\tau)$ and Matérn kernels, $\nu$ has a smoothing effect.
>
> Derivation:
> We use the notation introduced in Appendix F.1. Given a kernel function of the form $\nu k_h(\tau)$, we have $\alpha=(\nu K+\lambda I)^{-1}y=\frac{1}{\nu}(K+\frac{\lambda}{\nu}I)^{-1}y$. On the other hand, $f(x)=\nu k(x,\mathcal{X}')\alpha=\nu k(x,\mathcal{X}')\frac{1}{\nu}(K+\frac{\lambda}{\nu}I)^{-1}y=k(x,\mathcal{X}')(K+\lambda' I)^{-1}y$ which is similar to what we have in Appendix F.1 except for the scaled regularization weight, $\lambda':=\frac{\lambda}{\nu}$.
>
> **"Unclear connection to SIREN“:**
>
> This is a very good observation, thank you. It’s true, Theorem 3 applies to all stationary kernel functions. This is a direct consequence of Bochner’s theorem, which states that stationary kernels and their spectral density are Fourier duals; it follows that every stationary kernel function has a spectral density. We have added a note to the paper (L338-340) and Appendix D (L778-782). We believe that this observation has made Theorem 3 even stronger, serving as a powerful tool to study the connection between widely used SIRENs and *any* stationary kernel function (although Matérn kernels already unify most of the nowadays used stationary kernels).
>
> If you feel we still did not properly address any of your concerns, please let us know. Thank you again for your efforts to evaluate our manuscript.
>
> ---
>
> [1] Tancik et al. Fourier features let networks learn high frequency functions in low dimensional domains. NeurIPS, 2020.
>
> [2] Rahimi et al. Random features for large-scale kernel machines. NeurIPS, 2007.
>
> [3] Yaglom. Correlation Theory of Stationary and Related Random Functions. Springer Series in Statistics, 1987.

---

> > ### Comment · Reviewer_JfMa · 2024-11-25
> >
> > Thank you to the authors for providing a detailed rebuttal. While some of my concerns have been adequately addressed, the following issues remain:
> >
> > “Hence, the reconstructed surface remains exactly the same, regardless of the input point cloud’s location in space (translating the point cloud does not affect the reconstructed surface).”
> >
> > What is the practical advantage of this compared to centering the input point cloud at the origin before reconstruction? In my view, the primary benefit seems to lie in ensuring a consistent reconstruction of local parts. I would appreciate further clarification on this point. The authors mention:
> > “The reconstructed surface of the chair legs in Figure 6 (now Figure 9) is not symmetrical because the input point cloud is not symmetrical.” Are there other examples that demonstrate this proposed advantage?
> >
> >
> > "...which is not stationary. Hence, Bochner’s theorem does not apply. Hence, Bochner’s theorem does not apply"
> >
> > Wouldn’t it be straightforward to address the case of the dot-product kernel by centering the input to achieve zero mean? Furthermore, the main point is that, intuitively, the spectral analysis seems more closely tied to the use of a local support kernel rather than any other specific property.
> >
> >
> > "From a practical perspective, however, in both variants, i.e., for a kernel of the form \nu k(\tau) and Matérn kernels,
> >  has a smoothing effect."
> >
> > With that in mind, what are the advantages of the Matérn kernel compared to $\tau \cdot \exp(-\tau^2/h)$?

---

> > > ### Author Response · Authors · 2024-11-28
> > >
> > > Dear Reviewer JfMa,
> > >
> > > we are glad to hear that we could resolve most of your concerns. We will now address your remaining concerns.
> > >
> > > **"What is the practical advantage of this compared to centering the input point cloud at the origin before reconstruction? In my view, the primary benefit seems to lie in ensuring a consistent reconstruction of local parts“:**
> > >
> > > We added an extensive discussion (including an experimental verification) about the implications of stationarity in the context of surface reconstruction to Appendix A, and also revised Section 3.1 completely. In essence, you are completely right; stationarity ensures
> > > 1. Translation invariance: No matter where the input points are located in space, a stationary kernel, such as the Matérn kernel, always yields the *same* surface reconstruction (this is because a stationary kernel’s value only depends on *relative* positions rather than the absolute location of the input points as the case for non-stationary kernels, such as the arc-cosine kernel).
> > > 2. Consistent reconstruction: Stationarity also ensures a consistent reconstruction of objects in a scene (or parts of an object) with *similar geometric properties* as long as the relative distances between points on each instance (or part) remain the same. In other words, if an object appears twice in a scene, it will be reconstructed similarly. This is in contrast to the arc-cosine kernel, whose value does depend on the absolute position of points. As we show in Figure 8 and detail in Appendix A, the arc-cosine kernel does not yield the same reconstructed surface for multiple instances of the same object in a scene.
> > >
> > > We honestly have to admit that we completely missed discussing the second point, and are more than happy that the reviewer pointed this out. We think that this greatly improved the paper, especially Section 3.1 (together with the discussion in Appendix A), and also brought to light another huge benefit of Matérn kernels. Thank you for bringing this up, and thank you for asking again.
> > >
> > > **"Wouldn’t it be straightforward to address the case of the dot-product kernel by centering the input to achieve zero mean?“:**
> > >
> > > The dot-product kernel, say $k(x,y)=x^\top y$, is non-stationary, therefore *not* translation invariant. However, it can be made translation invariant by subtracting the mean, i.e., a *shift invariant dot-product kernel* looks like this: $k_0(x,y)=(x-c)^\top(y-c)$, where $c$ is the mean (centroid) computed over all input points. Why does this work? Consider a constant translation, $t$, of all input points. If we translate the input point cloud, its centroid will also shift by $t$. Thus, $k_0(x+t,y+t)=(x+t-(c+t))^\top(y+t-(c+t))=(x-c)^\top(y-c)=k_0(x,y)$, which shows that $k_0$ is indeed translation invariant. The exact same argument works for the arc-cosine kernel as stated in Eq. (3), see Appendix A. However, just because a kernel is translation invariant *does not necessarily mean* it is stationary. Clearly, while $k_0$ is translation invariant, it is not (and will never be) stationary because it still depends on absolute positions, $x,y$ (it can not be written as a function of $x-y$—this is, however, a necessary condition for a kernel to be stationary; see Bochner’s theorem). The same argument again holds for the arc-cosine kernel.
> > >
> > > In conclusion, although the dot-product and arc-cosine kernel can be made translation invariant, there's generally no way to "make it stationary" (it remains non-stationary). Hence, Bochner's theorem can not be applied.

---

> ### Author Response · Authors · 2024-11-28
>
> **"With that in mind, what are the advantages of the Matérn kernel compared to $\tau\cdot\exp(-\tau^2/h)$?**
>
> This is a great question. However, we would like to kindly ask whether the reviewer refers to $\tau\exp(-\tau^2/h)$ or $\nu\exp(-\tau^2/h)$, as the second form has already been mentioned in an earlier comment.
>
> For now and based on the context, we will assume that the question is about the kernel $\nu\exp(-\tau^2/h)$. Please let us know if not, we’re happy to discuss the other form as well.
>
> While, from a *practical* perspective, $\nu$ has a smoothing effect in both variants, it plays a much "bigger role“ in the family of Matérn kernels (besides simply regulating smoothness). First off, we would like to mention for a general kernel of the form $\nu k(\tau)$, where $k$ is *any* stationary kernel, the parameter $\nu$ does *not* change the smoothness of the actual kernel *function*, $k$ (clearly, $\nu$ does not change the shape of the kernel function; it just scales its magnitude). This is different for Matérn kernels, where $\nu$ changes the smoothness properties of the actual kernel function (this can be easily visualized by drawing random functions from a Gaussian Process with Matérn kernels for different values of $\nu$). Specifically, $\nu$ determines the differentiability class of the kernel and, more importantly, allows direct and *explicit control* over it. This allows an easy and *intuitive* injection of inductive biases (such as smoothness assumptions) into the surface reconstruction problem, as described in Section 3.1, L182-188. Since $\nu$ does not change the actual shape of the kernel function for a kernel of the form $\nu k(\tau)$, it does not have any interpretation other than just being a "scaling factor“ (in fact, as shown in our previous comment, $\nu$ is not even a parameter of the kernel function itself—it is actually a parameter of the KRR problem).
>
> Considering the exact kernel the reviewer is asking about, $\nu\exp(-\tau^2/h)$ (i.e., $k(\tau)=\exp(-\tau^2/h$), we would like to add that this is simply a scaled version of the Gaussian kernel. Our experiments have shown that the Gaussian kernel is simply too smooth in practice, leading to bad surface reconstructions (see results presented in Tables 1 and 4).
>
> In conclusion, we believe the biggest advantage of Matérn kernels compared to $\nu\exp(-\tau^2/h)$ (which is, de-facto, a Matérn kernel) is that they are more flexible, allow an easier injection of inductive biases, and are generally more interpretable. This is because the parameter $\nu$ has a clear and intuitive meaning.
>
> We hope that this adequately addresses your concerns. However, if you feel we still did not properly address any of your concerns, please let us know. We are always happy to answer. Thank you again for your valuable feedback, which greatly helped us improve the paper.

---

> > ### Comment · Reviewer_JfMa · 2024-12-02
> >
> > Thank you for the additional clarifications. On a related note, if stationarity is the desired property, an alternative **kernel** to consider, aside from the learned dot product kernel discussed earlier, is $k(x,y) = (\phi_1(x-y))^T \phi_2(X) (\phi_2(x-y))$.
> > Here, $\phi_1$​ is a learned function that maps points to a high-dimensional feature space, while $\phi_2$ is a learned function that maps the input point cloud to a positive definite matrix.
> >
> > I've updated my score to 8.

---

> > > ### Author Response · Authors · 2024-12-03
> > >
> > > Dear Reviewer JfMa,
> > >
> > > thank you very much for increasing your score. We're glad we could clarify all of your concerns and would like to thank you again for your honest, fair, and constructive feedback.
> > >
> > > As a last note: The kernel you are proposing looks very interesting, and we appreciate your suggestion. Lifting *relative differences* to a higher-dimensional feature space instead of absolute positions seems like a nice idea. Without having analyzed this kernel in detail yet, probably one disadvantage of this form is that the function $\phi_1$ needs to be queried $n^2$ times (where $n$ is the number of input points), instead of only $n$ times when using the learned dot-product kernel discussed earlier. Nevertheless, this is definitely an interesting idea which we will think about in the future.

---

### Official Review · Reviewer_nQr2 · 2024-11-07

**Soundness:** 3
**Presentation:** 3
**Contribution:** 2
**Rating:** 5
**Confidence:** 3

**Summary:**

This paper enumerates the use of a class of kernels that can be constructed with two interpretable parameters: bandwidth and smoothness. Specific choices of these parameters lead to well-known kernels like Laplace and Gaussian kernels.
This paper elaborates both on theory and experimental evaluation. Specifically, one can derive certain expressions for bounds on the eigenvalue decay rate (EDR), and reconstruction error. More interestingly the authors also draw a connection to neural networks and show  that Matern kernels mimic the computation of two-layer SIRENs if the layer width approaches infinity.

On the practical side, a range of experiments is provided on surface reconstruction on ShapeNet and SRB (SHape Reconstruction Benchmark). The experiments show favorable metrics for certain fixed and learnable Matern Kernel choices. The comparison is drawn to prior methods like NKF, SPSR, and Neural Splines on challenging metrics like performance with noise, and convergence speed.

**Strengths:**

- Overall I found the paper compiled quite well and tells an story of how Matern Kernels can be interesting theoretically and practically useful. The exposition focussing on this class of kernels is quite comprehensive.
- I particularly liked the connection to SIRENS
- I found the experiments to be well made. Reconstructions and evaluations are indeed favorable in comparison to their competing methods

**Weaknesses:**

- Despite a well-made submission, I found it hard to appreciate a strong take-home message. A lot of the material on Matern Kernels appears to be well-known from statistics and I don't find it surprising that they can be tuned well in practice by modulating the bandwidth and smoothness parameters. In this sense, I am a bit disappointed about a direct novel result (other than the relation connecting to SIRENS)
- I did not fully understand and appreciate the excessive focus on bettering the arc-cosine kernel

**Questions:**

figure on page 6 needs labels and a quick caption

**Details Of Ethics Concerns:**

Overall this is a decent but not a competitive paper at ICLR. I don't have strong reasons to either reject the paper (because the exposition on introducing Matern Kernels is comprehensive) or champion it - because of the lack of overall novelty. I gave the benefit of my doubt to a weak acceptance.

---

> ### Author Response · Authors · 2024-11-21
>
> Dear Reviewer nQr2,
>
> thank you for the professional evaluation of our manuscript and your valuable comments. We tried our best to address your concerns and hope we could clarify them below and in the updated version of the paper. Please reach out to us if you still have open questions or points that need further clarification. We’re happy to answer them.
>
> **"I found it hard to appreciate a strong take-home message" and "I did not fully understand and appreciate the excessive focus on bettering the arc-cosine kernel":**
>
> Neural Splines (NS) [1] was the first paper showing that kernel methods are extremely competitive for 3D surface reconstruction, outperforming both, modern neural-network-based methods such as IGR, SIREN, and FFN, but also classical methods like Biharmonic RBF [2], SVR, and SPSR by a *large* margin. From a practical perspective, NS’ success was achieved by simply changing the kernel function within the framework introduced in [2]; instead of using an RBF, NS uses the first-order arc-cosine kernel. From a theoretical perspective, the authors of NS demonstrated that the first-order arc-cosine kernel mimics the computation of an infinite-wide two-layer ReLU network when bottom-layer weights are fixed and initialized from a Gaussian distribution. We take a similar route in this paper.
>
> The benefits of kernel methods over neural networks in the context of 3D reconstruction are listed on page 2 in [1]; they are a big motivation for this work.
>
> The problem with the arc-cosine kernel is that (1) it is not tunable because it does not have parameters (there is no way to improve surface reconstructions); (2) it is slow to compute due to its complexity, leading to long run times; (3) it is not translation invariant, making reconstructed surfaces dependent on the actual embedding of the point cloud; (4) being zonal, it is less interpretable and harder to analyze compared to *stationary* Matérn kernels (easy to visualize, known spectral density, better understood theoretically).
>
> From a practical perspective, the goal of this paper is to show that a simple change of the kernel function from arc-cosine to Matérn yields (1) more controllability over the surface reconstruction problem as it allows users to tune their reconstructions, overall leading to increased reconstruction quality; (2) a significant speedup while not affecting reconstruction accuracy. We could achieve both, see Section 4. To theoretically justify our choice for the Matérn kernel family, we show in Theorem 3 that Matérn kernels can be identified with the NTK of SIREN networks—similar to the NS paper [1], in which authors derived a relation between ReLU networks and the arc-cosine kernel. To put Matérn kernels even further into context, we derive a relation between both kernel functions in Corollary 5. Finally, we also analyze how and why "tuning" in theory works and give insights into the selection of the tunable parameter, $h$, by inspecting the reconstruction error as a function of $h$ (Section 3.2 in the updated version). These insights are highly valuable for users in practice, leading to a better understanding and *interpretation* of surface reconstruction results.
>
> TL;DR: To summarize, we would phrase a practical take-home message as follows: Simply changing the kernel function in the NS framework to Matérn kernels leads to significantly faster surface reconstructions with higher accuracy. Surface reconstructions can be tuned by the user by varying an easy-to-understand and intuitive parameter, $h$. Compared to the arc-cosine kernel, the implementation of Matern kernels is straightforward and requires essentially two lines of standard PyTorch code. To make Matern kernels data-depended, one could either use the NKSR framework which is highly sophisticated, well-engineered, and challenging to implement, or simply exchange the kernel function in the NKF framework which leads to approximately the same reconstruction quality as NKSR while having a five times shorter training time. To increase the reconstruction quality and convergence speed (thus reducing training time) of the original NKF framework, one simply has to change the kernel function.
>
> To make the take-home message clearer, we added an explicit list of contributions to the introduction, see L115-121. We also updated L122-128.
>
> **"Figure on page 6 needs labels and a quick caption“:**
>
> We added a caption to the figure on page 6. Additionally, we noticed a mistake (five graphs, only four labels) in this figure and corrected it. Thank you!
>
> If you feel we still did not properly address any of your concerns, please let us know. Thank you again for your efforts to evaluate our manuscript.
>
> ---
> [1] Williams et al. Neural Splines: Fitting 3D Surfaces with Inifinitely-Wide Neural Networks. CVPR, 2021.
>
> [2] Carr et al. Surface interpolation with radial basis functions for medical imaging. IEEE TMI, 1997.

---

### Author Response · Authors · 2024-12-04

With the discussion phase coming to an end, we would like to thank again all reviewers for their honest and highly valuable comments and suggestions. Based on the reviewer’s feedback, we believe that we successfully addressed all of their concerns.

In accordance with the reviewer’s comments, we extensively revised our manuscript, with the key edits being:
- We added a paragraph in L114-119 that explicitly and concisely states our contributions, and completely revised L300-312 which now better connects our theoretical analysis to its practical use.

- We completely revised Section 3.1, now better highlighting Matérn kernel’s benefits, and added an extensive discussion to Appendix A.

- We added a comparison to four state-of-the-art methods on the challenging SRB in Table 1.

- We re-rendered almost all figures to include sampled input points and ground truth surfaces.

- We removed the former Section 3 and merged relevant parts of it with Section 4 (now Section 3).

- We added a new experiment in which we reconstruct high-frequency textures, presenting qualitative and quantitative evaluations and two datasets to Section 4.2. We also carried out a sampling irregularity experiment according to suggestions by Reviewer QSTk (see comments for results). We will add these results to the paper once possible.

We were delighted to see that, so far, two out of four reviewers raised their score.

---

### Meta-Review · Area_Chair_Zx5A · 2024-12-20

**Metareview:**

This paper suggests Matern Kernels for applications of implicit surface representation and reconstruction from oriented point clouds (i.e., point clouds with normals). The Matern Kernels are used, similar to other kernel functions, to build (via linear combinations) implicit functions representing the signed distance function, the zero level of which represents the surface of interest. The significance in this particular choice of kernels is in the tunable parameters that provide a family including known kernels, such as the Gaussian kernel, and provide refined control over smoothness of the reconstructed surface and the reconstruction error. Experiments with these kernels for various surface reconstruction benchmarks demonstrate the efficacy. On the down side, some reviewers feel this paper presents a rather straightforward application of a known family of kernels in a standard way which diminishes from its contribution a bit.

**Additional Comments On Reviewer Discussion:**

No additional comments.

---

### Decision · Program_Chairs · 2025-01-22

Accept (Poster)